# DESSERT: An Efficient Algorithm for Vector Set Search with Vector Set Queries

**Joshua Engels**
ThirdAI
josh.adam.engels@gmail.com

**Benjamin Coleman**
ThirdAI
benjamin.ray.coleman@gmail.com

**Vihan Lakshman**
ThirdAI
vihan@thirdai.com

**Anshumali Shrivastava**
ThirdAI, Rice University
anshu@thirdai.com, anshumali@rice.edu

## Abstract

We study the problem of *vector set search* with *vector set queries*. This task is analogous to traditional near-neighbor search, with the exception that both the query and each element in the collection are *sets* of vectors. We identify this problem as a core subroutine for semantic search applications and find that existing solutions are unacceptably slow. Towards this end, we present a new approximate search algorithm, DESSERT (**D**ESSERT **E**ffeciently **S**earches **S**ets of **E**mbeddings via **R**etrieval **T**ables). DESSERT is a general tool with strong theoretical guarantees and excellent empirical performance. When we integrate DESSERT into ColBERT, a state-of-the-art semantic search model, we find a 2-5x speedup on the MS MARCO and LoTTE retrieval benchmarks with minimal loss in recall, underscoring the effectiveness and practical applicability of our proposal.

## 1 Introduction

Similarity search is a fundamental driver of performance for many high-profile machine learning applications. Examples include web search [16], product recommendation [33], image search [21], de-duplication of web indexes [29] and friend recommendation for social media networks [39]. In this paper, we study a variation on the traditional vector search problem where the dataset $D$ consists of a collection of *vector sets* $D = \{S_1, ... S_N\}$ and the query $Q$ is also a vector set. We call this problem *vector set search* with *vector set queries* because both the collection elements and the query are sets of vectors. Unlike traditional vector search, this problem currently lacks a satisfactory solution.

Furthermore, efficiently solving the vector set search problem has immediate practical implications. Most notably, the popular ColBERT model, a state-of-the-art neural architecture for semantic search over documents [23], achieves breakthrough performance on retrieval tasks by representing each query and document as a set of BERT token embeddings. ColBERT's current implementation of vector set search over these document sets, while superior to brute force, is prohibitively slow for real-time inference applications like e-commerce that enforce strict search latencies under 20-30 milliseconds [34, 5]. Thus, a more efficient algorithm for searching over sets of vectors would have significant implications in making state-of-the-art semantic search methods feasible to deploy in large-scale production settings, particularly on cost-effective CPU hardware.

Given ColBERT's success in using vector sets to represent documents more accurately, and the prevailing focus on traditional single-vector near-neighbor search in the literature [1, 41, 19, 12, 14, 28, 18], we believe that the potential for searching over sets of representations remains largely untapped. An efficient algorithmic solution to this problem could enable new applications in domains

37th Conference on Neural Information Processing Systems (NeurIPS 2023).

where multi-vector representations are more suitable. To that end, we propose DESSERT, a novel randomized algorithm for efficient set vector search with vector set queries. We also provide a general theoretical framework for analyzing DESSERT and evaluate its performance on standard passage ranking benchmarks, achieving a 2-5x speedup over an optimized ColBERT implementation on several passage retrieval tasks.

## 1.1 Problem Statement

More formally, we consider the following problem statement.

**Definition 1.1.** Given a collection of $N$ vector sets $D = \{S_1, ...S_N\}$, a query set $Q$, a failure probability $\delta \geq 0$, and a set-to-set relevance score function $F(Q, S)$, the *Vector Set Search Problem* is the task of returning $S^*$ with probability at least $1 - \delta$:

$$S^\star = \underset{i \in \{1,...N\}}{\operatorname{argmax}} F(Q, S_i)$$

Here, each set $S_i = \{x_1, ...x_{m_i}\}$ contains $m_i$ vectors with each $x_j \in \mathbb{R}^d$, and similarly $Q = \{q_1, ...q_{m_q}\}$ contains $m_q$ vectors with each $q_j \in \mathbb{R}^d$.

We further restrict our consideration to structured forms of $F(Q, S)$, where the relevance score consists of two "set aggregation" or "variadic" functions. The inner aggregation $\sigma$ operates on the pairwise similarities between a single vector from the query set and each vector from the target set. Because there are $|S|$ elements in $S$ over which to perform the aggregation, $\sigma$ takes $|S|$ arguments. The outer aggregation $A$ operates over the $|Q|$ scores obtained by applying $A$ to each query vector $q \in Q$. Thus, we have that

$$F(Q, S) = A(\{Inner_{q,S} : q \in Q\})$$
$$Inner_{q,S} = \sigma(\{\text{sim}(q, x) : x \in S\})$$

Here, sim is a vector similarity function. Because the inner aggregation is often a maximum or other non-linearity, we use $\sigma(\cdot)$ to denote it, and similarly since the outer aggregation is often a linear function we denote it with $A(\cdot)$. These structured forms for $F = A \circ \sigma$ are a good measure of set similarity when they are monotonically non-decreasing with respect to the similarity between any pair of vectors from $Q$ and $S$.

## 1.2 Why is near-neighbor search insufficient?

It may at first seem that we could solve the Vector Set Search Problem by placing all of the individual vectors into a near-neighbor index, along with metadata indicating the set to which they belonged. One could then then identify high-scoring sets by finding near neighbors to each $q \in Q$ and returning their corresponding sets.

There are two problems with this approach. The first problem is that a single high-similarity interaction between $q \in Q$ and $x \in S$ does not imply that $F(Q, S)$ will be large. For a concrete example, suppose that we are dealing with sets of word embeddings and that $Q$ is a phrase where one of the items is "keyword." With a standard near-neighbor index, $Q$ will match (with 100% similarity) any set $S$ that also contains "keyword," regardless of whether the other words in $S$ bear any relevance to the other words in $Q$. The second problem is that the search must be conducted over all individual vectors, leading to a search problem that is potentially very large. For example, if our sets are documents consisting of roughly a thousand words and we wish to search over a million documents, we now have to solve a billion-scale similarity search problem.

**Contributions:** In this work, we formulate and carefully study the set of vector search problem with the goal of developing a more scalable algorithm capable of tackling large-scale semantic retrieval problems involving sets of embeddings. Specifically, our research contributions can be summarized as follows:

1. We develop the first non-trivial algorithm, DESSERT, for the vector set search problem that scales to large collections ($n > 10^6$) of sets with $m > 3$ items.
2. We formalize the vector set search problem in a rigorous theoretical framework, and we provide strong guarantees for a common (and difficult) instantiation of the problem.

3. We provide an open-source C++ implementation of our proposed algorithm that has been deployed in a real-world production setting[1]. Our implementation scales to hundreds of millions of vectors and is 3-5x faster than existing approximate set of vector search techniques. We also describe the implementation details and tricks we discovered to achieve these speedups and provide empirical latency and recall results on passage retrieval tasks.

## 2   Related Work

**Near-Neighbor Search:**   Near-neighbor search has received heightened interest in recent years with the advent of vector-based representation learning. In particular, considerable research has gone into developing more efficient *approximate* near-neighbor (ANN) search methods that trade off an exact solution for sublinear query times. A number of ANN algorithms have been proposed, including those based on locality-sensitive hashing [1, 41], quantization and space partition methods [19, 12, 14], and graph-based methods [28, 18]. Among these classes of techniques, our proposed DESSERT framework aligns most closely with the locality-sensitive hashing paradigm. However, nearly all of the well-known and effective ANN methods focus on searching over individual vectors; our work studies the search problem for sets of entities. This modification changes the nature of the problem considerably, particularly with regards to the choice of similarity metrics between entities.

**Vector Set Search:**   The general problem of vector set search has been relatively understudied in the literature. A recent work on database lineage tracking [25] addresses this precise problem, but with severe limitations. The proposed approximate algorithm designs a concatenation scheme for the vectors in a given set, and then performs approximate search over these concatenated vectors. The biggest drawback to this method is scalability, as the size of the concatenated vectors scales quadratically with the size of the vector set. This leads to increased query latency as well as substantial memory overhead; in fact, we are unable to apply the method to the datasets in this paper without terabytes of RAM. In this work, we demonstrate that DESSERT can scale to thousands of items per set with a linear increase (and a slight logarithmic overhead) in query time, which, to our knowledge, has not been previously demonstrated in the literature.

**Document Retrieval:**   In the problem of document retrieval, we receive queries and must return the relevant documents from a preindexed corpus. Early document retrieval methods treated each documents as bags of words and had at their core an inverted index [30]. More recent methods embed each document into a single representative vector, embed the query into the same space, and performed ANN search on those vectors. These semantic methods achieve far greater accuracies than their lexical predecessors, but require similarity search instead of inverted index lookups [15, 33, 26].

**ColBERT and PLAID:**   ColBERT [23] is a recent state of the art algorithm for document retrieval that takes a subtly different approach. Instead of generating a single vector per document, ColBERT generates a *set* of vectors for each document, approximately one vector per word. To rank a query, ColBERT also embeds the query into a set of vectors, filters the indexed sets, and then performs a brute force *sum of max similarities* operation between the query set and each of the document sets. ColBERT's passage ranking system is an instantiation of our framework, where $\text{sim}(q, x)$ is the cosine similarity between vectors, $\sigma$ is the max operation, and $A$ is the sum operation.

In a similar spirit to our work, PLAID [37] is a recently optimized form of ColBERT that includes more efficient filtering techniques and faster quantization based set similarity kernels. However, we note that these techniques are heuristics that do not come with theoretical guarantees and do not immediately generalize to other notions of vector similarity, which is a key property of the theoretical framework behind DESSERT.

## 3   Algorithm

At a high level, a DESSERT index $\mathcal{D}$ compresses the collection of target sets into a form that makes set to set similarity operations efficient to calculate. This is done by replacing each set $S_i$ with a sketch $\mathcal{D}[i]$ that contains the LSH values of each $x_j \in S_i$. At query time, we compare the corresponding

---

[1]https://github.com/ThirdAIResearch/Dessert

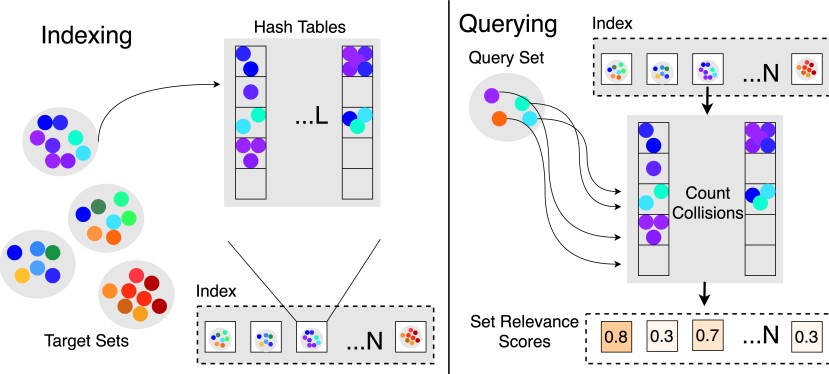

Figure 1: The DESSERT indexing and querying algorithms. During indexing (left), we represent each target set as a set of hash values ($L$ hashes for each element). To query the index (right), we approximate the similarity between each target and query element by averaging the number of hash collisions. These similarities are used to approximate the set relevance score for each target set.

LSH values of the query set $Q$ with the hashes in each $\mathcal{D}[i]$ to approximate the pairwise similarity matrix between $Q$ and $S$ (Figure 1). This matrix is used as the input for the aggregation functions $A$ and $\sigma$ to rank the target sets and return an estimate of $S^*$.

We assume the existence of a locality-sensitive hashing (LSH) family $\mathcal{H} \subset (\mathbb{R}^d \rightarrow \mathbb{Z})$ such that for all LSH functions $h \in \mathcal{H}$, $p(h(x) = h(y)) = \text{sim}(x, y)$. LSH functions with this property exist for cosine similarity (signed random projections) [8], Euclidean similarity ($p$-stable projections) [11], and Jaccard similarity (minhash or simhash) [6]. LSH is a well-developed theoretical framework with a wide variety of results and extensions [4, 3, 20, 40]. See Appendix C for a deeper overview.

Algorithm 1 describes how to construct a DESSERT index $\mathcal{D}$. We first take $L$ LSH functions $f_t$ for $t \in [1, L]$, $f_t \in \mathcal{H}$. We next loop over each $S_i$ to construct $\mathcal{D}[i]$. For a given $S_i$, we arbitrarily assign an identifier $j$ to each vector $x \in S_i$, $j \in [1, |S_i|]$. We next partition the set $[1, m_i]$ using each hash function $h_t$, such that for a partition $p_t$, indices $j_1$ and $j_2$ are in the same set in the partition iff $h(S_{j_1}) = h(S_{j_2})$. We represent the results of these partitions in a universal hash table indexed by hash function id and hash function value, such that $\mathcal{D}[i]_{t,h} = \{j \mid x_j \in S_i \wedge f_t(x_j) = h\}$.

Algorithm 2 describes how to query a DESSERT index $\mathcal{D}$. At a high level, we query each sketch $\mathcal{D}_i$ to get an estimate of $F(Q, S_i)$, $score_i$, and then take the argmax over the *estimates* to get an estimate of $\text{argmax}_{i \in \{1,...N\}} F(Q, S_i)$. To get these estimates, we first compute the hashes $h_{t,q}$ for each query $q$ and LSH function $f_t$. Then, to get an estimate $score_i$ for a set $S_i$, we loop over the hashes $h_{t,q}$ for each query vector $q$ and count how often each index $j$ appears in $\mathcal{D}[i]_{t,h_{t,q}}$. After we finish this step, we have a count for each $j$ that represents how many times $h_t(q) = h_t(x_j)$. Equivalently, since $p(h(x) = h(y)) = \text{sim}(x, y)$, if we divide by $L$ we have an estimate for $\text{sim}(x_j, q)$. We then apply $\sigma$ to these estimates and save the result in a variable $\text{agg}_q$ to build up the inputs to $A$, and then apply $A$ to get our final estimate for $F(Q, S_i)$, which we store in $score_i$.

## 4 Theory

In this section, we analyze DESSERT's query runtime and provide probabilistic bounds on the correctness of its search results. We begin by finding the hyperparameter values and conditions that are necessary for DESSERT to return the top-ranked set with high probability. Then, we use these results to prove bounds on the query time. In the interest of space, we defer proofs to the Appendix.

**Notation:** For the sake of simplicity of presentation, we suppose that all target sets have the same number of elements $m$, i.e. $|S_i| = m$. If this is not the case, one may replace $m_i$ with $m_{\max}$ in our analysis. We will use the boldface vector $\mathbf{s}(q, S_i) \in \mathbb{R}^{|S_i|}$ to refer to the set of pairwise similarity calculations $\{\text{sim}(q, x_1), \ldots, \text{sim}(q, x_{m_i})\}$ between a query vector and the elements of $S_i$, and we will drop the subscript $(q, S_i)$ when the context is clear. See Table 1 for a complete notation reference.

Table 1: Notation table with examples from the document search application.

| Notation | Definition | Intuition (Document Search) |
|---|---|---|
| $D$ | Set of target vector sets | Collection of documents |
| $N$ | Cardinality $|D|$ | Number of documents |
| $\mathcal{D}$ | DESSERT index of $D$ | Search index data structure |
| $S_i$ | Target vector set $i$ | $i$th document |
| $Q$ | Query vector set | Multi-word query (e.g., a question) |
| $S^*$ | See Definition 1.1 | The most relevant document to $Q$ |
| $x_j \in S_i$ | $j$th vector in target set $S_i$ | Embedding from document $i$ |
| $q_j \in Q$ | $j$th vector in query set $Q$ | Embedding from a query |
| $d$ | $s_j, x_j \in \mathbb{R}^d$ | Embedding dimension |
| $m_i, m$ | Cardinality $|S_i|$, $m_i = m$ | Number of embeddings in $i$th document |
| $F(Q, S_i)$ | $Q$ and $S_i$ relevance score | Measures query-document similarity |
| $score_i$ | Estimate of $F(Q, S_i)$ | Approximation of relevance score |
| $\mathcal{D}[i]$ | Sketch of $i$th target set | Estimates relevance score for $S_i$ and any $Q$ |
| $\text{sim}(a, b)$ | $a$ and $b$ vector similarity | Embedding similarity |
| $A, \sigma$ | See Section 1.1 | Components of relevance score |
| $L$ | Number of hashes | Larger $L$ increases accuracy and latency |
| $f_i$ | $i$th LSH function | Often maps nearby points to the same value |
| $\mathbf{s}(q, S_i), \mathbf{s}$ | $\text{sim}(q, x_j)$ for $x_j \in S_i$ | Query embedding similarities with $S_i$ |

---

**Algorithm 1** Building a DESSERT Index

1: **Input:** $N$ sets $S_i$, $|S_i| = m_i$
2: **Output:** A DESSERT index $\mathcal{D}$
3: $\mathcal{D}$ = an array of $N$ hash tables, each indexed by $x \in \mathbb{Z}^2$
4: **for** $i = 1$ to $N$ **do**
5:   **for** $x_j$ **in** $S_i$ **do**
6:     **for** $t = 1$ **to** $L$ **do**
7:       $h = f_t(x_j)$
8:       $\mathcal{D}[i]_{t,h} = \mathcal{D}[i]_{t,h} \cup \{j\}$
9: Return $\mathcal{D}$

---

**Algorithm 2** Querying a DESSERT Index

1: **Input:** DESSERT index $\mathcal{D}$, query set $Q$, $|Q| = m_q$.
2: **Output:** Estimate of $\text{argmax}_{i \in \{1...N\}} A \circ \sigma(Q, S_i)$
3: **for** $q$ **in** $Q$ **do**
4:   $h_{1,q}, \ldots, h_{L,q} = f_1(q), \ldots, f_L(q)$
5: **for** $i = 1$ to $N$ **do**
6:   **for** $q$ **in** $Q$ **do**
7:     $\hat{\mathbf{s}} = 0$
8:     **for** $t = 1$ **to** $L$ **do**
9:       **for** $j$ **in** $\mathcal{D}[i]_{t,h_{t,q}}$ **do**
10:         $\hat{\mathbf{s}}_j = \hat{\mathbf{s}}_j + 1$
11:     $\hat{\mathbf{s}} = \hat{\mathbf{s}} / \text{L}$
12:     $agg_q = \sigma(\hat{\mathbf{s}})$
13:   $score_i = A(\{agg_q \,|\, q \in Q\})$
14: Return $\text{argmax}_{i \in \{1...N\}} score_i$

---

## 4.1 Inner Aggregation

We begin by introducing a condition on the $\sigma$ component of the relevance score that allows us to prove useful statements about the retrieval process.

**Definition 4.1.** A function $\sigma(\mathbf{x}) : \mathbb{R}^m \to \mathbb{R}$ is $(\alpha, \beta)$-maximal on $U \subset \mathbb{R}^m$ if for $0 < \beta \leq 1 \leq \alpha$, $\forall x \in U$:

$$\beta \max \mathbf{x} \leq \sigma(\mathbf{x}) \leq \alpha \max \mathbf{x}$$

The function $\sigma(\mathbf{x}) = \max \mathbf{x}$ is a trivial example of an $(\alpha, \beta)$-maximal function on $\mathbb{R}^m$, with $\beta = \alpha = 1$. However, we can show that other functions also satisfy this definition:

**Lemma 4.1.1.** *If $\varphi(x) : \mathbb{R} \to \mathbb{R}$ is $(\alpha, \beta)$-maximal on an interval $I$, then the following function $\sigma(x) : \mathbb{R}^m \to \mathbb{R}$ is $\left(\alpha, \frac{\beta}{m}\right)$-maximal on $U = I^m$:*

$$\sigma(\mathbf{x}) = \frac{1}{m} \sum_{i=1}^{m} \varphi(x_i)$$

Note that in $\mathbb{R}$, the $(\alpha, \beta)$-maximal condition is equivalent to lower and upper bounds by linear functions $\beta x$ and $\alpha x$ respectively, so many natural functions satisfy Lemma 4.1.1. We are particularly interested in the case $I = [0, 1]$, and note that possible such $\varphi$ include $\varphi(x) = x$ with $\beta = \alpha = 1$, the exponential function $\varphi(x) = e^x - 1$ with $\beta = 1$, $\alpha = e - 1$, and the debiased sigmoid function $\varphi(x) = \frac{1}{1+e^{-x}} - \frac{1}{2}$ with $\beta \approx 0.23$, $\alpha = 0.25$. Our analysis of DESSERT holds when the $\sigma$ component of the relevance score is an $(\alpha, \beta)$ maximal function.

In line 12 of Algorithm 2, we estimate $\sigma(\mathbf{s})$ by applying $\sigma$ to a vector of normalized counts $\hat{\mathbf{s}}$. In Lemma 4.1.2, we bound the probability that a low-similarity set (one for which $\sigma(\mathbf{s})$ is low) scores well enough to outrank a high-similarity set. In Lemma 4.1.3, we bound the probability that a high-similarity set scores poorly enough to be outranked by other sets. Note that the failure rate in both lemmas decays exponentially with the number of hash tables $L$.

**Lemma 4.1.2.** *Assume $\sigma$ is $(\alpha, \beta)$-maximal. Let $0 < s_{\max} < 1$ be the maximum similarity between a query vector and the vectors in the target set and let $\hat{\mathbf{s}}$ be the set of estimated similarity scores. Given a threshold $\alpha s_{\max} < \tau < \alpha$, we write $\Delta = \tau - \alpha s_{\max}$, and we have*

$$\Pr[\sigma(\hat{\mathbf{s}}) \geq \alpha s_{\max} + \Delta] \leq m\gamma^L$$

*for $\gamma = \left( \frac{s_{\max}(\alpha - \tau)}{\tau(1 - s_{\max})} \right)^{\frac{\tau}{\alpha}} \left( \frac{\alpha(1 - s_{\max})}{\alpha - \tau} \right) \in (s_{\max}, 1)$. Furthermore, this expression for $\gamma$ is increasing in $s_{\max}$ and decreasing in $\tau$, and $\gamma$ has one sided limits $\lim_{\tau \searrow \alpha s_{\max}} \gamma = 1$ and $\lim_{\tau \nearrow \alpha} \gamma = s_{\max}$.*

**Lemma 4.1.3.** *With the same assumptions as Lemma 4.1.2 and given $\Delta > 0$, we have:*

$$Pr[\sigma(\hat{\mathbf{s}}) \leq \beta s_{\max} - \Delta] \leq 2e^{-2L\Delta^2/\beta^2}$$

## 4.2 Outer Aggregation

Our goal in this section is to use the bounds established previously to prove that our algorithm correctly ranks sets according to $F(Q, S)$. To do this, we must find conditions under which the algorithm successfully identifies $S^\star$ based on the approximate $F(Q, S)$ scores.

Recall that $F(Q, S)$ consists of two aggregations: the inner aggregation $\sigma$ (analyzed in Section 4.1) and the outer aggregation $A$. We consider normalized *linear* functions for $A$, where we are given a set of weights $0 \leq w \leq 1$ and we rank the target set according to a weighted linear combination of $\sigma$ scores.

$$F(Q, S) = \frac{1}{m} \sum_{j=1}^{m} w_j \sigma(\hat{\mathbf{s}}(q_j, S))$$

With this instantiation of the vector set search problem, we will proceed in Theorem 4.2 to identify a choice of the number of hash tables $L$ that allows us to provide a probabilistic guarantee that the algorithm's query operation succeeds. We will then use this parameter selection to bound the runtime of the query operation in Theorem 4.3.

**Theorem 4.2.** *Let $S^\star$ be the set with the maximum $F(Q, S)$ and let $S_i$ be any other set. Let $B^\star$ and $B_i$ be the following sums (which are lower and upper bounds for $F(Q, S^\star)$ and $F(Q, S_i)$, respectively)*

$$B^\star = \frac{\beta}{m_q} \sum_{j=1}^{m_q} w_j s_{\max}(q_j, S^\star) \qquad B_i = \frac{\alpha}{m_q} \sum_{j=1}^{m_q} w_j s_{\max}(q_j, S_i)$$

*Here, $s_{\max}(q, S)$ is the maximum similarity between a query vector $q$ and any element of the target set $S$. Let $B'$ be the maximum value of $B_i$ over any set $S_i \neq S$. Let $\Delta$ be the following value (proportional to the difference between the lower and upper bounds)*

$$\Delta = (B^\star - B')/3$$

*If $\Delta > 0$, a DESSERT structure with the following value[2] of $L$ solves the search problem from Definition 1.1 with probability $1 - \delta$.*

$$L = O\left( \log\left( \frac{Nm_q m}{\delta} \right) \right)$$

---

[2]$L$ additionally depends on the data-dependent parameter $\Delta$, which we elide in the asymptotic bound; see the proof in the appendix for the full expression for $L$.

### 4.3 Runtime Analysis

**Theorem 4.3.** *Suppose that each hash function call runs in time $O(d)$ and that $|\mathcal{D}[i]_{t,h}| < T \; \forall i, t, h$ for some positive threshold $T$, which we treat as a data-dependent constant in our analysis. Then, using the assumptions and value of L from Theorem 4.2, Algorithm 2 solves the Vector Set Search Problem in query time*

$$O\left(m_q \log(Nm_q m/\delta)d + m_q N \log(Nm_q m/\delta)\right)$$

This bound is an improvement over a brute force search of $O(m_q m N d)$ when $m$ or $d$ is large. The above theorem relies upon the choice of $L$ that we derived in Theorem 4.2.

## 5 Implementation Details

**Filtering:** We find that for large $N$ it is useful to have an initial lossy filtering step that can cheaply reduce the total number of sets we consider with a low false-negative rate. We use an inverted index on the documents for this filtering step.

To build the inverted index, we first perform $k$-means clustering on a representative sample of individual item vectors at the start of indexing. The inverted index we will build is a map from centroid ids to document ids. As we add each set $S_i$ to $\mathcal{D}$ in Algorithm 1, we also add it into the inverted index: we find the closest centroid to each vector $x \in S_i$, and then we add the document id $i$ to all of the buckets in the inverted index corresponding to those centroids.

This method is similar to PLAID, the recent optimized ColBERT implementation [37], but our query process is much simpler. During querying, we query the inverted index buckets corresponding to the closest *filter_probe* centroids to each query vector. We aggregate the buckets to get a count for each document id, and then only rank the *filter_k* documents with DESSERT that have the highest count.

**Space Optimized Sketches:** DESSERT has two features that constrain the underlying hash table implementation: (1) every document is represented by a hash table, so the tables must be low memory, and (2) each query performs many table lookups, so the lookup operation must be fast. If (1) is not met, then we cannot fit the index into memory. If (2) is not met, then the similarity approximation for the inner aggregation step will be far too slow. Initially, we tried a naive implementation of the table, backed by a std::vector, std::map, or std::unordered_map. In each case, the resulting structure did not meet our criteria, so we developed TinyTable, a compact hash table that optimizes memory usage while preserving fast access times. TinyTables sacrifice $O(1)$ update-access (which DESSERT does not require) for a considerable improvement to (1) and (2).

A TinyTable replaces the universal hash table in Algorithm 1, so it must provide a way to map pairs of (hash value, hash table id) to lists of vector ids. At a high level, a TinyTable is composed of $L$ inverted indices from LSH values to vector ids. Bucket $b$ of table $i$ consists of vectors $x_j$ such that $h_i(x_j) = b$. During a query, we simply need to go to the $L$ buckets that correspond to the query vector's $L$ lsh values to find the ids of $S_i$'s colliding vectors. This design solves (1), the fast lookup requirement, because we can immediately go to the relevant bucket once we have a query's hash value. However, there is a large overhead in storing a resizable vector in every bucket. Even an empty bucket will use $3 * 8 = 24$ bytes. This adds up: let $r$ be the hash range of the LSH functions (the number of buckets in the inverted index for each of the $L$ tables). If $N = 1M$, $L = 64$, and $r = 128$, we will use $N \cdot L \cdot r \cdot 24 = 196$ gigabytes even when *all of the buckets are empty*.

Thus, a TinyTable has more optimizations that make it space efficient. Each of the $L$ hash table repetitions in a TinyTable are conceptually split into two parts: a list of offsets and a list of vector ids. The vector ids are the concatenated contents of the buckets of the table with no space in between (thus, they are always some permutation of 0 through $m$ - 1). The offset list describes where one bucket ends and the next begins: the $i$th entry in the offset list is the (inclusive) index of the start of the $i$th hash bucket within the vector id list, and the $i + 1$th entry is the (exclusive) end of the $i$th hash bucket (if a bucket is empty, $indices[i] = indices[i+1]$). To save more bytes, we can further concatenate the $L$ offset lists together and the $L$ vector id lists together, since their lengths are always $r$ and $m_i$ respectively. Finally, we note that if $m \leq 256$, we can store all of the the offsets and ids can be safely be stored as single byte integers. Using the same hypothetical numbers as before, a *filled* TinyTable with $m = 100$ will take up just $N(24 + L(m + r + 1)) = 14.7$GB.

**The Concatenation Trick:** In our theory, we assumed LSH functions such that $p(h(x) = h(y)) = \text{sim}(x, y)$. However, for practical problems such functions lead to overfull buckets; for example, GLOVE has an average vector cosine similarity of around $0.3$, which would mean each bucket in the LSH table would contain a third of the set. The standard trick to get around this problem is to *concatenate* $C$ hashes for each of the $L$ tables together such that $p(h(x) = h(y)) = \text{sim}(x, y)^C$. Rewriting, we have that

$$\text{sim}(x, y) = \exp\left(\frac{\ln\left[p(h(x) = h(y))\right]}{C}\right) \tag{1}$$

During a query, we count the number of collisions across the $L$ tables and divide by $L$ to get $\hat{p}(h(x) = h(y))$ on line 11 of Algorithm 2. We now additionally pass $count/L$ into Equation 1 to get an accurate similarity estimate to pass into $\sigma$ on line 12. Furthermore, evaluating Equation 1 for every collision probability estimate is slow in practice. There are only $L + 1$ possible values for the $count/L$, so we precompute the mapping in a lookup table.

## 6 Experiments

**Datasets:** We tested DESSERT on both synthetic data and real-world problems. We first examined a series of synthetic datasets to measure DESSERT's speedup over a reasonable CPU brute force algorithm (using the PyTorch library [35] for matrix multiplications). For this experiment, we leave out the prefiltering optimization described in Section 5 to better show how DESSERT performs on its own. Following the authors of [25], our synthetic dataset consists of random groups of Glove [36] vectors; we vary the set size $m$ and keep the total number of sets $N = 1000$.

We next experimented with the MS MARCO passage ranking dataset (Creative Commons License) [32], $N \approx 8.8M$. The task for MS MARCO is to retrieve passages from the corpus relevant to a query. We used ColBERT to map the words from each passage and query to sets of embedding vectors suitable for DESSERT [37]. Following [37], we use the development set for our experiments, which contains 6980 queries.

Finally, we computed the full resource-accuracy tradeoff for ten of the LoTTE out-of-domain benchmark datasets, introduced by ColBERTv2 [38]. We excluded the pooled dataset, which is simply the individual datasets merged.

**Experiment Setup:** We ran our experiments on an Intel(R) Xeon(R) CPU E5-2680 v3 machine with 252 GB of RAM. We restricted all experiments to 4 cores (8 threads). We ran each experiment with the chosen hyperparameters and reported overall average recall and average query latency. For all experiments we used the average of max similarities scoring function.

### 6.1 Synthetic Data

The goal of our synthetic data experiment was to examine DESSERT's speedup over brute force vector set scoring. Thus, we generated synthetic data where both DESSERT and the brute force implementation achieved perfect recall so we could compare the two methods solely on query time.

The two optimized brute force implementations we tried both used PyTorch, and differed only in whether they computed the score between the query set and each document set individually ("Individual") or between the query set and all document sets at once using PyTorch's highly performant reduce and reshape operations ("Combined").

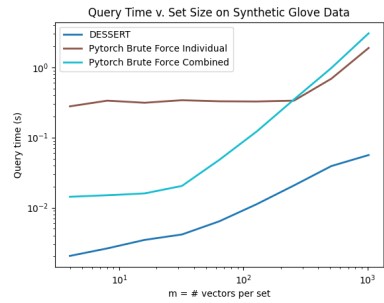

Figure 2: Query time for DESSERT vs. brute force on 1000 random sets of $m$ glove vectors with the $y$-axis as a log scale. Lower is better.

In each synthetic experiment, we inserted 1000 documents of size $m$ for $m \in [2, 4, 8, 16, ..., 1024]$ into DESSERT and the

brute force index. The queries in each experiment were simply the documents with added noise. The DESSERT hyperparameters we chose were $L = 8$ and $C = \log_2(m) + 1$. The results of our experiment, which show the relative speedup of using DESSERT at different values of $m$, are in Figure 2. We observe that DESSERT achieves a consistent 10-50x speedup over the optimized Pytorch brute force method and that the speedup increases with larger $m$ (we could not run experiments with even larger $m$ because the PyTorch runs did not finish within the time allotted).

## 6.2 Passage Retrieval

Passage retrieval refers to the task of identifying and returning the most relevant passages from a large corpus of documents in response to a search query. In these experiments, we compared DESSERT to PLAID, ColBERT's heavily-optimzed state-of-the-art late interaction search algorithm, on the MS MARCO and LoTTE passage retrieval tasks.

We found that the best ColBERT hyperparameters were the same as reported in the PLAID paper, and we successfully replicated their results. Although PLAID offers a way to trade off time for accuracy, this tradeoff only increases accuracy at the sake of time, and even then only by a fraction of a percent. Thus, our results represent points on the recall vs time Pareto frontier that PLAID cannot reach.

**MS MARCO Results**  For MS MARCO, we performed a grid search over DESSERT parameters $C = \{4, 5, 6, 7\}$, $L = \{16, 32, 64\}$, $filter\_probe = \{1, 2, 4, 8\}$, and $filter\_k = \{1000, 2048, 4096, 8192, 16384\}$. We reran the best configurations to obtain the results in Table 2. We report two types of results: methods tuned to return $k = 10$ results and methods tuned to return $k = 1000$ results. For each, we report DESSERT results from a low latency and a high latency part of the Pareto frontier. For $k = 1000$ we use the standard $R@1000$ metric, the average recall of the top 1 passage in the first 1000 returned passages. This metric is meaningful because retrieval pipelines frequently rerank candidates after an initial retrieval stage. For $k = 10$ we use the standard $MRR@10$ metric, the average mean reciprocal rank of the top 1 passage in the first 10 returned passages. Overall, DESSERT is 2-5x faster than PLAID with only a few percent loss in recall.

| Method | Latency (ms) | $MRR@10$ | Method | Latency (ms) | $R@1000$ |
|--------|--------------|----------|--------|--------------|----------|
| DESSERT | 9.5 | $35.7 \pm 1.14$ | DESSERT | 22.7 | $95.1 \pm 0.49$ |
| DESSERT | 15.5 | $37.2 \pm 1.14$ | DESSERT | 32.3 | $96.0 \pm 0.45$ |
| PLAID | 45.1 | $39.2 \pm 1.15$ | PLAID | 100 | $97.5 \pm 0.36$ |

Table 2: MS MARCO passage retrieval, with methods optimized for k=10 (left) and k=1000 (right). Intervals denote 95% confidence intervals for average latency and recall.

**LoTTE Results**  For LoTTE, we performed a grid search over $C = \{4, 6, 8\}$, $L = \{32, 64, 128\}$, $filter\_probe = \{1, 2, 4\}$, and $filter\_k = \{1000, 2048, 4096, 8192\}$. In Figure 3, we plot the full Pareto tradeoff for DESSERT on the 10 LoTTE datasets (each of the 5 categories has a "forum" and "search" split) over these hyperparameters, as well as the single lowest-latency point achievable by PLAID. For all test datasets, DESSERT provides a Pareto frontier that allows a tradeoff between recall and latency. For both Lifestyle test splits, both Technology test splits, and the Recreation and Science test-search splits, DESSERT achieves a 2-5x speedup with minimal loss in accuracy. On Technology, DESSERT even exceeds the accuracy of PLAID at half of PLAID's latency.

## 7 Discussion

We observe a substantial speedup when we integrate DESSERT into ColBERT, even when compared against the highly-optimized PLAID implementation. While the use of our algorithm incurs a slight recall penalty – as is the case with most algorithms that use randomization to achieve acceleration – Table 2 and Figure 3 shows that we are Pareto-optimal when compared with baseline approaches.

We are not aware of any algorithm other than DESSERT that is capable of latencies in this range for set-to-set similarity search. While systems such as PLAID are tunable, we were unable to get them to operate in this range. For this reason, DESSERT is likely the only set-to-set similarity search algorithm that can be run in real-time production environments with strict latency constraints.

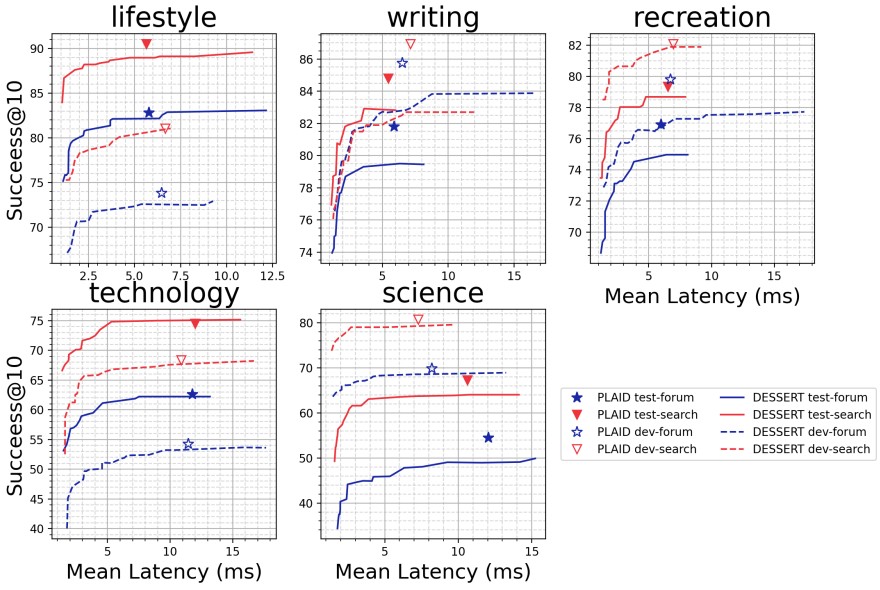

Figure 3: Full Pareto frontier of DESSERT on the LoTTE datasets. The PLAID baseline shows the lowest-latency result attainable by PLAID (with a FAISS-IVF base index and centroid pre-filtering).

We also ran a single-vector search baseline using ScaNN, the leading approximate kNN index [14]. ScaNN yielded 0.77 Recall@1000, substantially below the state of the art. This result reinforces our discussion in Section 1.2 on why single-vector search is insufficient.

**Broader Impacts and Limitations:** Ranking and retrieval are important steps in language modeling applications, some of which have recently come under increased scrutiny. However, our algorithm is unlikely to have negative broader effects, as it mainly enables faster, more cost-effective search over larger vector collections and does not contribute to the problematic capabilities of the aforementioned language models. Due to computational limitations, we conduct our experiments on a relatively small set of benchmarks; a larger-scale evaluation would strengthen our argument. Finally, we assume sufficiently high relevance scores and large gaps in our theoretical analysis to identify the correct results. These hardness assumptions are standard for LSH.

## 8  Conclusion

In this paper, we consider the problem of vector set search with vector set queries, a task understudied in the existing literature. We present a formal definition of the problem and provide a motivating application in semantic search, where a more efficient algorithm would provide considerable immediate impact in accelerating late interaction search methods. To address the large latencies inherent in existing vector search methods, we propose a novel randomized algorithm called DESSERT that achieves significant speedups over baseline techniques. We also analyze DESSERT theoretically and, under natural assumptions, prove rigorous guarantees on the algorithm's failure probability and runtime. Finally, we provide an open-source and highly performant C++ implementation of our proposed DESSERT algorithm that achieves 2-5x speedup over ColBERT-PLAID on the MS MARCO and LoTTE retrieval benchmarks. We also note that a general-purpose algorithmic framework for vector set search with vector set queries could have impact in a number of other applications, such as image similarity search [42], market basket analysis [22], and graph neural networks [43], where it might be more natural to model entities via sets of vectors as opposed to restricting representations to a single embedding. We believe that DESSERT could provide a viable algorithmic engine for enabling such applications and we hope to study these potential use cases in the future.

## 9 Acknowledgments

This work was completed while the authors were working at ThirdAI. We do not have any external funding sources to acknowledge.

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

# A  Proofs of Main Results

**Lemma 4.1.1.** *If $\varphi(x) : \mathbb{R} \to \mathbb{R}$ is $(\alpha, \beta)$-maximal on an interval $I$, then the following function $\sigma(x) : \mathbb{R}^m \to \mathbb{R}$ is $\left(\alpha, \frac{\beta}{m}\right)$-maximal on $U = I^m$:*

$$\sigma(\mathbf{x}) = \frac{1}{m} \sum_{i=1}^{m} \varphi(x_i)$$

*Proof.* Take some $\mathbf{x} \in D$ (so each $x_i \in I$). Since in $\mathbb{R}$, $\max(x) = x$, we have from the definition of $(\alpha, \beta)$-maximal that

$$\beta x \leq x \leq \alpha x$$

For the upper bound, we have

$$\sigma(\mathbf{x}) = \frac{1}{m} \sum_{i=1}^{m} \varphi(x_i) \leq \frac{1}{m} \sum_{i=1}^{m} \alpha x_i = \frac{\alpha}{m} \sum_{i=1}^{m} x_i \leq \frac{\alpha}{m} (m \max(\mathbf{x})) = \alpha \max(\mathbf{x})$$

where the second inequality follows by the properties of the max function.

For the lower bound, we have that

$$\sigma(\mathbf{x}) = \frac{1}{m} \sum_{i=1}^{m} \varphi(x_i) \geq \frac{1}{m} \sum_{i=1}^{m} \beta x_i = \frac{\beta}{m} \sum_{i=1}^{m} x_i \geq \frac{\beta}{m} \max \mathbf{x}$$

where the the second inequality again follows by the properties of the max function. □

**Lemma 4.1.2.** *Assume $\sigma$ is $(\alpha, \beta)$-maximal. Let $0 < s_{\max} < 1$ be the maximum similarity between a query vector and the vectors in the target set and let $\hat{\mathbf{s}}$ be the set of estimated similarity scores. Given a threshold $\alpha s_{\max} < \tau < \alpha$, we write $\Delta = \tau - \alpha s_{\max}$, and we have*

$$\Pr[\sigma(\hat{\mathbf{s}}) \geq \alpha s_{\max} + \Delta] \leq m\gamma^L$$

*for $\gamma = \left(\frac{s_{\max}(\alpha-\tau)}{\tau(1-s_{\max})}\right)^{\frac{\tau}{\alpha}} \left(\frac{\alpha(1-s_{\max})}{\alpha-\tau}\right) \in (s_{\max}, 1)$. Furthermore, this expression for $\gamma$ is increasing in $s_{\max}$ and decreasing in $\tau$, and $\gamma$ has one sided limits $\lim_{\tau \searrow \alpha s_{\max}} \gamma = 1$ and $\lim_{\tau \nearrow \alpha} \gamma = s_{\max}$.*

*Proof.* We first apply a generic Chernoff bound to $\sigma(\hat{\mathbf{s}})$, which gives us the following bounds for any $t > 0$:

$$\Pr[\sigma(\hat{\mathbf{s}}) \geq \tau] = \Pr[e^{t\sigma(\hat{\mathbf{s}})} \geq e^{t\tau}] \leq \frac{\mathbb{E}[e^{t\sigma(\hat{\mathbf{s}})}]}{e^{t\tau}}$$

We now proceed by continuing to bound the numerator. Because $\sigma$ is $(\alpha, \beta)$-maximal, we can bound $\sigma(\hat{\mathbf{s}})$ with $\alpha \max \hat{\mathbf{s}}$. We can further bound $\max \hat{\mathbf{s}}$ by bounding the maximum with the sum and the sum with $m$ times the maximal element. We are now left with the formula for the moment generating function for $\hat{s}_{\max}$. $\hat{s}_{\max} \sim$ scaled binomial $L^{-1}\mathcal{B}(s_{\max}, L)$, so we can directly substitute the binomial moment generating function into the expression:

$$\mathbb{E}[e^{t\sigma(\hat{\mathbf{s}})}] \leq \mathbb{E}[e^{t\alpha \max_j \hat{s}_j}] \leq \sum_{j=1}^{m_i} \mathbb{E}[e^{t\alpha \hat{s}_j}] \leq m\mathbb{E}[e^{t\alpha \hat{s}_{\max}}]$$

$$= m(1 - s_{\max} + s_{\max}e^{\frac{\alpha t}{L}})^L$$

Combining these two equations yields the following bound:

$$\Pr[\sigma(\hat{\mathbf{s}}) \geq \tau] \leq me^{-t\tau}(1 - s_{\max} + s_{\max}e^{\frac{\alpha t}{L}})^L$$

We wish to select a value of $t$ to minimize the upper bound. By setting the derivative of the upper bound to zero, and imposing $0 < \tau < \alpha$, $\alpha \geq 1$, and $0 < s_{max} < 1$, we find that

$$t^\star = \frac{L}{\alpha} \ln\left(\frac{\tau(1 - s_{\max})}{s_{\max}(\alpha - \tau)}\right)$$

This is greater than zero when the numerator inside the $\ln$ is greater than the denominator, or equivalently when $\tau > s_{\max}\alpha$. Thus the valid range for $\tau$ is $(s_{\max}\alpha, \alpha)$ (and similarly the valid range for $s_{\max}$ is $(0, \tau/\alpha)$). These bounds have a natural interpretation: to be meaningful, the threshold must be between the expected value and the maximum value for $\alpha$ times a $p = s_{\max}$ binomial. Substituting $t = t^\star$ into our upper bound, we obtain:

$$\Pr[\sigma(\hat{\mathbf{s}}) \geq \tau] \leq m\left(\left(\frac{\tau(1 - s_{\max})}{s_{\max}(\alpha - \tau)}\right)^{-\frac{\tau}{\alpha}}\left(\frac{\alpha(1 - s_{\max})}{\alpha - \tau}\right)\right)^L$$

Thus we have that

$$\gamma = \left(\frac{s_{\max}(\alpha - \tau)}{\tau(1 - s_{\max})}\right)^{\frac{\tau}{\alpha}}\left(\frac{\alpha(1 - s_{\max})}{\alpha - \tau}\right)$$

We will now prove our claims about $\gamma$ viewed as a function of $s_{\max} \in (0, \tau/\alpha)$ and $\tau \in (s_{\max}\alpha, \alpha)$. We will first examine the limits of $\gamma$ with respect to $\tau$ at the ends of its range. Since $\gamma$ is continuous, we can find one of the limits by direct substitution:

$$\lim_{\tau \searrow s_{\max}\alpha} \gamma = \lim_{s_{\max} \nearrow \tau/\alpha} \gamma = \left(\frac{\tau/\alpha(\alpha - \tau)}{\tau(1 - \tau/\alpha)}\right)^{\frac{\tau}{\alpha}}\left(\frac{\alpha(1 - \tau/\alpha)}{\alpha - \tau}\right) = 1^{\frac{\tau}{\alpha}} * 1 = 1$$

The second limit is harder; we merge $\gamma$ into one exponent and then simplify:

$$\lim_{\tau \nearrow \alpha} \gamma = \lim_{\tau \nearrow \alpha}\left(\frac{s_{\max}(\alpha - \tau)^{1 - \alpha}\,{}^{\tau}\alpha^{\alpha/\tau}}{\tau(1 - s_{\max})^{1 - \alpha/\tau}}\right)^{\frac{\tau}{\alpha}}$$

$$= \lim_{\tau \nearrow \alpha}\frac{s_{\max}(\alpha - \tau)^{1 - \alpha}\,{}^{\tau}\alpha^{\alpha/\tau}}{\tau(1 - s_{\max})^{1 - \alpha/\tau}}$$

$$= \lim_{\tau \nearrow \alpha} s_{\max}(\alpha - \tau)^{1 - \alpha/\tau} = \lim_{\tau \nearrow \alpha}\left(s_{\max}(\alpha - \tau)^{\alpha - \tau}\right)^{-1/\tau}$$

$$= s_{\max}\left(\lim_{\alpha - \tau \searrow 0}\left((\alpha - \tau)^{\alpha - \tau}\right)\right)^{-1/\alpha}$$

$$= s_{\max}(1)^{-1/\alpha} = s_{\max}$$

where we use the fact that $\lim_{x \searrow 0} x^x = e^{\lim_{x \searrow 0} x \ln(x)} = 1$ (we can see that $\lim_{x \to 0} x \ln(x) = \lim_{x \to 0} \ln(x)/(1/x) = 0$ with L'Hopital's rule). We next find the partial derivatives of $\gamma$:

$$\frac{\delta\gamma}{\delta s_{\max}} = \frac{(\tau - \alpha s_{\max})\left(\frac{\alpha s_{\max} - s_{\max}\tau}{\tau - s_{\max}\tau}\right)^{\frac{\tau}{\alpha}}}{s_{\max}(\alpha - \tau)} \qquad \frac{\delta\gamma}{\delta\tau} = \frac{(s_{\max} - 1)\left(\frac{\alpha s_{\max} - s_{\max}\tau}{\tau - s_{\max}\tau}\right)^{\frac{\tau}{\alpha}}\ln\left(\frac{\tau - s_{\max}\tau}{\alpha s_{\max} - s_{\max}\tau}\right)}{\alpha - \tau}$$

We are interested in the signs of these partial derivatives. First examining $\frac{\delta\gamma}{\delta s_{\max}}$, $\tau > \alpha s_{\max} \implies \tau - \alpha s_{\max} > 0$. Similarly, $\alpha > \tau \implies \alpha - \tau > 0$ and $s_{\max}(\alpha - \tau) = s_{\max}\alpha - s_{\max}\tau > 0$. Finally, $s_{\max} < 1 \implies \tau(1 - s_{\max}) = \tau - \tau s_{\max} > 0$. Thus every term is positive and the entire fraction is positive. Next examining $\frac{\delta\gamma}{\delta\tau}$, by similar logic $\alpha - \tau > 0$ and $\tau - s_{\max}\tau > 0$ and $\alpha s_{\max} - s_{\max}\tau > 0$. For the $\ln$, since $\tau > \alpha s_{\max}$, $\tau - s_{\max}\tau > \alpha s_{\max} - s_{\max}\tau$, so the numerator is greater than the denominator and the $\ln$ is positive. Finally, since $s_{\max} < 1$, $s_{\max} - 1 < 0$, and thus the entire fraction has a single negative term in the product, so it is negative.

This completes our lemma: $\gamma$ is a strictly decreasing function of $\tau$ and a strictly increasing function of $s_{\max}$. Since $\tau$ is decreasing and has a leftward limit of 1 and a rightward limit of $s_{\max}$, all values for $\gamma$ are in $(s_{\max}, 1)$.

First, we will make a substitution. We note that $\gamma$ is a strictly decreasing function on this interval of $\tau$ with range $(s_{\max}, 1)$. To see this, we will first make the following change of variabls:

$$\tau = \frac{\alpha(k+s)}{k+1}$$

for $k \in (0, \infty)$. This parameterizes $\tau \in (s_{\max}\alpha, \alpha)$ as a weighted sum of $s_{\max}\alpha$ and $\alpha$. Plugging in and simplifying, we have that

$$\gamma = \left( \frac{s_{\max}}{k + s_{\max}} \right)^{\frac{k+s_{\max}}{k+1}} (k+1)$$

This is a continous function over $k \in (0, \infty)$ and $s_{\max} \in ()$

$\square$

**Lemma 4.1.3.** *With the same assumptions as Lemma 4.1.2 and given $\Delta > 0$, we have:*

$$Pr[\sigma(\hat{\mathbf{s}}) \leq \beta s_{\max} - \Delta] \leq 2e^{-2L\Delta^2/\beta^2}$$

*Proof.* We will prove this lemma with a chain of inequalities, starting with $\Pr[\sigma(\hat{\mathbf{s}}) \leq \beta s_{\max} - \Delta]$:

$$
\begin{aligned}
\Pr[\sigma(\hat{\mathbf{s}}) \leq \beta s_{\max} - \Delta] &\leq \Pr[\beta \max \hat{\mathbf{s}} \leq \beta s_{\max} - \Delta] \\
&\leq \Pr[\beta \hat{s_{\max}} \leq \beta s_{\max} - \Delta] \\
&= \Pr[\beta s_{\max} - \beta \hat{s_{\max}} \geq \Delta] \\
&\leq \Pr[|\beta s_{\max} - \beta \hat{s_{\max}}| \geq \Delta] = \Pr[|\beta \hat{s_{\max}} - \beta s_{\max}| \geq \Delta] \\
&\leq 2e^{-2L\Delta^2/\beta^2}
\end{aligned}
$$

The explanations for each step are as follows:

1. Because $\sigma(\hat{\mathbf{s}}) \geq \beta \max \mathbf{s}$, we can replace $\sigma(\hat{\mathbf{s}})$ with $\beta \max \mathbf{s}$ and the probability will be strictly larger.

2. By the definition of the max operator, each individual $\hat{s}_i \leq \max \hat{\mathbf{s}}$, and in particular this is true for $\hat{s_{\max}}$ (the estimated similarity for the ground-truth maximum similarity vector). Thus, we have $\beta \hat{s_{\max}} \leq \beta \max \hat{\mathbf{s}}$, so we can again apply a replacement to get a further upper bound.

3. Rearranging.

4. Because $Pr[|a - b| \geq c] = Pr[a - b \geq c] + Pr[b - a \geq c]$

5. $\hat{s}_{\max}$ is the sum of $L$ Bernoulli trials with success probability $s_{\max}$ and scaled by $\beta/L$. Thus, we can directly apply the Hoeffding ineuqliaty with $L$ trials with success probability $\frac{\beta s_{\max}}{L}$.

$\square$

**Theorem 4.2.** *Let $S^\star$ be the set with the maximum $F(Q, S)$ and let $S_i$ be any other set. Let $B^\star$ and $B_i$ be the following sums (which are lower and upper bounds for $F(Q, S^\star)$ and $F(Q, S_i)$, respectively)*

$$B^\star = \frac{\beta}{m_q} \sum_{j=1}^{m_q} w_j s_{\max}(q_j, S^\star) \qquad B_i = \frac{\alpha}{m_q} \sum_{j=1}^{m_q} w_j s_{\max}(q_j, S_i)$$

*Here, $s_{\max}(q, S)$ is the maximum similarity between a query vector $q$ and any element of the target set $S$. Let $B'$ be the maximum value of $B_i$ over any set $S_i \neq S$. Let $\Delta$ be the following value (proportional to the difference between the lower and upper bounds)*

$$\Delta = (B^\star - B')/3$$

*If $\Delta > 0$, a DESSERT structure with the following value[3] of $L$ solves the search problem from Definition 1.1 with probability $1 - \delta$.*

$$L = O\left(\log\left(\frac{Nm_q m}{\delta}\right)\right)$$

*Proof.* For set $S^\star$ to have the highest estimated score $\hat{F}(Q, S^\star)$, we need all other sets to have lower scores. Our overall proof strategy will find a minimum $L$ that upper bounds the probability that each inner aggregation of a set $S \neq S^\star$ is greater than $\Delta + \alpha s'_{j,\max}$ and a minimum $L$ that lower bounds the probability that the inner aggregation of $S^*$ is less $\beta s^*_{j,\max} - \Delta$. Finally, we will show that an $L$ that is a maximum of these two values solves the search problem.

**Upper Bound**: We start with the upper bound on $S_i \neq S^*$: we have from Lemma 4.1.2 that

$$\Pr[\sigma(\hat{\mathbf{s}}_\mathbf{i}, q_j) \geq \alpha s_{i,\max} + \Delta] \leq m\gamma_i^L$$

with

$$\gamma_i = \left(\frac{(\Delta + \alpha s_{i,\max})(1 - s_{i,\max})}{s_{i,\max}(\alpha - (\Delta + \alpha s_{i,\max}))}\right)^{-\frac{\Delta + \alpha s_{i,\max}}{\alpha}} \left(\frac{\alpha(1 - s_{i,\max})}{\alpha - (\Delta + \alpha s_{i,\max})}\right)$$

and $\gamma_i \in (0, 1)$. To make our analysis simpler, we are interested in the maximum $\gamma_{\max}$ of all these $\gamma_i$ as a function of $\Delta$, since then all of these bounds will hold with the same $\gamma$, making it easy to solve for $L$. Since $\lim_{s_{\max} \searrow 0} \gamma = 0$ and $\lim_{s_{\max} \nearrow 1 - \Delta} = 1 - \Delta/\alpha$, there must be some $\gamma_{\max} \in (1 - \Delta/\alpha, 1)$ that maximizes this expression over any $s_{\max}$. This exact maximum is hard to find analytically, but we are guaranteed that it is less than 1 by Lemma 4.1.2. We will use the term $\gamma_{\max}$ in our analysis, since it is data dependent and guaranteed to be in the range $(0, 1)$. We also numerically plot some values of $\gamma_{\max}$ here with $\alpha = 1$ to give some intuition for what the function looks like over different $\Delta$; we note that it is decreasing in $\Delta$ and approximates a linear function for $\Delta >> 0$.

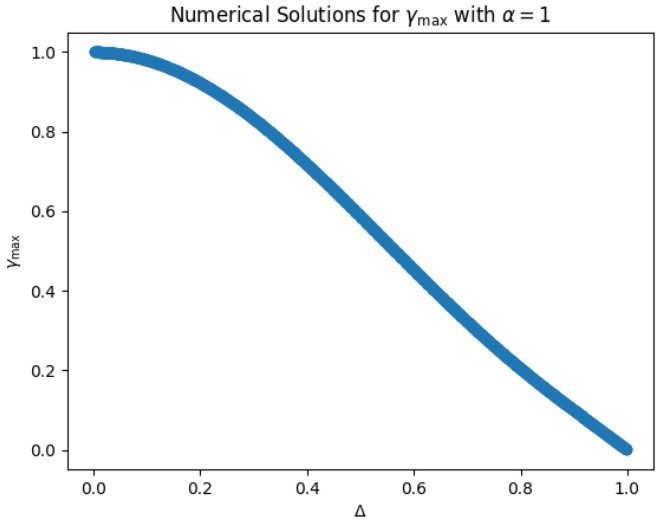

To hold with the union bound over all $N - 1$ target sets and all $m_q$ query vectors with probability $\frac{\delta}{2}$, we want the probability that our bound holds on a single set and query vector to be less than $\frac{\delta}{2(N-1)m_q}$. We find that this is true with $L \geq \log\frac{2(N-1)m_q m}{\delta} \log(\frac{1}{\gamma_{\max}})^{-1}$ for any $q_j$ and $S_i$:

$$\Pr[\sigma(\hat{\mathbf{s}}_\mathbf{i}, q_j) \geq \alpha s_{i,\max} + \Delta] \leq m\gamma_i^L \leq m\gamma_{\max}^L \leq m\left(\gamma_{\max}\right)^{\log\frac{2(N-1)m_q m}{\delta} \log(\frac{1}{\gamma_{\max}})^{-1}} = \frac{\delta}{2(N-1)m_q}$$

---

[3]$L$ additionally depends on the data-dependent parameter $\Delta$, which we elide in the asymptotic bound; see the proof in the appendix for the full expression for $L$.

**Lower Bound** We next examine the lower bound on $S^*$: we have from Lemma 4.1.3 that

$$\Pr[\sigma(\hat{\mathbf{s}}^*, q_j) \leq \beta s_{*,\max} - \Delta] \leq 2e^{-2L\Delta^2/\beta^2}$$

To hold with the union bound over all $m_q$ query vectors with probability $\frac{\delta}{2}$, we want the probability that our bound holds on a single set and query vector to be less than $\frac{\delta}{2m_q}$. We find that this is true with $L \geq \frac{\log(\frac{4m_q}{\delta})\beta^2}{2\Delta^2}$ for any $q_j$:

$$\Pr[\sigma(\hat{\mathbf{s}}^*, q_j) \leq \beta s_{*,\max} - \Delta] \leq 2e^{-2L\Delta^2/\beta^2} \leq 2e^{-2\frac{\log(\frac{4m_q}{\delta})\beta^2}{2\Delta^2}\Delta^2/\beta^2} = \frac{\delta}{2m_q}$$

**Putting it Together**

Let

$$L = \max\left(\frac{\log\frac{2(N-1)m_q m}{\delta}}{\log(\frac{1}{\gamma_{\max}})}, \frac{\log(\frac{4m_q}{\delta})\beta^2}{2\Delta^2}\right)$$

Then the upper and lower bounds we derived in the last two sections both apply. Let $\mathbb{1}$ be the random variable that is 1 when the $m * m_q * (N-1)$ upper bounds and the $m_q$ lower bounds hold and that is 0 otherwise. Consider all sets $S_i \neq S^*$. Then the probability we solve the Vector Set Search Problem from Definition 1.1 is equal to the probability that all $\forall_i, (\hat{F}(Q, S^*) - \hat{F}(Q, S_i) > 0)$. We now lower bound this probability:

$$\Pr\left(\forall_i(\hat{F}(Q, S^*) - \hat{F}(Q, S_i) > 0)\right)$$

$$= \Pr\left(\forall_i \left(\frac{1}{m_q}\sum_{j=1}^{m_q} w_j\sigma(\hat{\mathbf{s}}^*, q_j) - \frac{1}{m_q}\sum_{j=1}^{m_q} w_j\sigma(\hat{\mathbf{s}}_i, q_j) > 0\right)\right) \qquad \text{Definition of } \hat{F}$$

$$= \Pr\left(\forall_i \left(\sum_{j=1}^{m_q} w_j(\sigma(\hat{\mathbf{s}}^*, q_j) - \sigma(\hat{\mathbf{s}}_i, q_j)) > 0\right)\right)$$

$$= \Pr\left(\forall_i \left(\sum_{j=1}^{m_q} w_j(\sigma(\hat{\mathbf{s}}^*, q_j) - \sigma(\hat{\mathbf{s}}_i, q_j)) > 0 \middle| \mathbb{1} = 1\right)\right)\Pr(\mathbb{1} = 1) \qquad \Pr(A) \geq \Pr(A \wedge B)$$

$$\geq \Pr\left(\forall_i \left(\sum_{j=1}^{m_q} w_j(\beta s_{*,\max} - \Delta - (\alpha s_{i,\max} + \Delta)) > 0\right)\right)\Pr(\mathbb{1} = 1) \qquad \text{Bounds hold on } \mathbb{1} = 1$$

$$= \Pr\left(\forall_i \left(\sum_{j=1}^{m_q} w_j(\beta s_{*,\max} - \alpha s_{i,\max}) > 2\Delta\sum_{j=1}^{m_q} w_j\right)\right)\Pr(\mathbb{1} = 1)$$

$$= \Pr\left(\forall_i \left(m_q(B^* - B_i) > 2\Delta\sum_{j=1}^{m_q} w_j\right)\right)\Pr(\mathbb{1} = 1) \qquad \text{Definition of } B^*, B_i$$

$$\geq \Pr\left(\forall_i \left(m_q(B^* - B') > 2\Delta\sum_{j=1}^{m_q} w_j\right)\right)\Pr(\mathbb{1} = 1) \qquad \text{Definition of } B'$$

$$\geq \Pr\left(\forall_i \left(3m_q\Delta > 2\Delta\sum_{j=1}^{m_q} w_j\right)\right)\Pr(\mathbb{1} = 1) \qquad \text{Definition of } \Delta$$

$$\geq \Pr\left(\forall_i (3m_q\Delta > 2m_q\Delta)\right)\Pr(\mathbb{1} = 1) \qquad w_j \leq 1$$

$$= 1 * (\mathbb{1} = 1) \qquad \Delta > 0$$

$$= 1 - (\mathbb{1} = 0)$$

$$\geq 1 - (m * m_q * (N-1) * \frac{\delta}{2(N-1)m_q} + \frac{\delta}{2m_q} * m_q) = 1 - \delta \qquad \text{Union bound}$$

and thus DESSERT solves the Vector Set Search Problem with this choice of $L$. Finally, we can now examine the expression for $L$ to determine its asymptotic behavior. Dropping the positive data dependent constants $\frac{1}{\gamma_{\max}}$, $\frac{1}{2\Delta^2}$, and $\beta^2$, the left term in the $\max$ for $L$ is $O(\log(\frac{Nm_q m}{\delta}))$ and the right term in the $\max$ is $O(\log(\frac{m_q}{\delta}))$, and thus $L = O\left(\log(\frac{Nm_q m}{\delta})\right)$.

$\square$

**Theorem 4.3.** *Suppose that each hash function call runs in time $O(d)$ and that $|\mathcal{D}[i]_{t,h}| < T \ \forall i,t,h$ for some positive threshold $T$, which we treat as a data-dependent constant in our analysis. Then, using the assumptions and value of L from Theorem 4.2, Algorithm 2 solves the Vector Set Search Problem in query time*

$$O\left(m_q \log(Nm_q m/\delta)d + m_q N \log(Nm_q m/\delta)\right)$$

*Proof.* If we suppose that each call to the hash function $f_t$ is $O(d)$, the runtime of the algorithm is

$$O\left(nLd + \sum_{i=0}^{n-1} \sum_{k=0}^{N-1} \sum_{t=0}^{L-1} |M_{k,t,f_t(q_j)}|\right)$$

To bound this quantity, we use the sparsity assumption we made in the theorem: no set $S_i$ contains too many elements that are very similar to a single query vector $q_j$. Formally, we require that

$$|\mathcal{D}[i]_{t,h}| < T \quad \forall i,t,h$$

for some positive threshold $T$. With this assumption, the runtime of Algorithm 2 is

$$O\left(m_q L d + m_q N L T\right)$$

Plugging in the $L$ we found in our previous theorem, and treating $T$ as data dependent constant, we have that the runtime of Algorithm 2 is

$$O\left(m_q \log(Nm_q m/\delta)d + m_q N \log(Nm_q m/\delta)\right)$$

which completes the proof.

$\square$

# B  Hyperparameter Settings

Settings for DESSERT corresponding to the first row in the left part of Table 2, where DESSERT was optimized for returning 10 documents in a low latency part of the Pareto frontier:

```
hashes_per_table (C) = 7
num_tables (L) = 32
filter_k = 4096
filter_probe = 1
```

Settings for DESSERT corresponding to the second row in the left part of Table 2, where DESSERT was optimized for returning 10 documents in a high latency part of the Pareto frontier:

```
hashes_per_table (C) = 7
num_tables (L) = 64
filter_k = 4096
filter_probe = 2
```

Settings for DESSERT corresponding to the first row in the right part of Table 2, where DESSERT was optimized for returning 1000 documents in a low latency part of the Pareto frontier:

```
hashes_per_table (C)= 6
num_tables (L) = 32
filter_k = 8192
filter_probe = 4
```

Settings for DESSERT corresponding to the second row in the right part of Table 2, where DESSERT was optimized for returning 1000 documents in a high latency part of the Pareto frontier:

```
hashes_per_table (C) = 7
num_tables (L) = 32
filter_k = 16384
filter_probe = 4
```

Intuitively, these parameter settings make sense: increase the initial filtering size and the number of total hashes for higher accuracy, and increase the initial filtering size for returning more documents (1000 vs. 10).

## C    Background on Locality-Sensitive Hashing and Inverted Indices for Similarity Search

Here, we offer a refresher on using locality-sensitive hashing for similarity search with a basic inverted index structure.

Consider a set of distinct vectors $X = \{x_1, \ldots, x_N\}$ where each $x_i \in \mathbb{R}^d$. A hash function $h$ with a range $m$ maps each $x_i$ to an integer in the range $[1, m]$. Two vectors $x_i$ and $x_j$ are said to "collide" when $h(x_i) = h(x_j)$.

As a warmup, we will first consider the case of a hash function $h$ drawn from a set of universal hash functions $H$. Under such a function, if $i \neq j$, $p(h(x) = h(y)) = \frac{1}{m}$; such families exist in practice [7]. We can build an inverted index using this hash function by mapping each hash value in $[1, m]$ to the set of vectors $X_v$ that have this hash value. Then, given a new vector $y$, we can query the inverted index with $v = h(y)$. We can see that $y \in X$ iff $y \in X_v$. Such an index is in a sense solving a search problem, if we only care about finding exact duplicates of our search query. Additionally, we can solve the nearest neighbor problem with this index in time $O(N)$, by going to every bucket and checking the distance of a query against every vector in the bucket.

Now, in a similar way as in the universal case, let $h$ to be drawn from a family of locality-sensitive hash functions $H$. At a high level, instead of mapping vectors uniformly to $[1, m]$, $h$ maps vectors that are close together to the same hash value more often. Formally, if we define a "close" threshold $r_1$, a "far" threshold $r_2$, a "close" probability $p_1$, and a "far" probability $p_2$, with $p_1 > p_2$ and $r_1 < r_2$, then we say $H$ is $(r_1, r_2, p_1, p_2)$-sensitive if

$$d(x, y) < r_1 \implies Pr(h(x) = h(y)) > p_1$$
$$d(x, y) > r_2 \implies Pr(h(x) = h(y)) < p_2$$

where $d$ is a distance metric. See [17] for the origin of locality-sensitive hashing and this definition. Intuitively, if we build an inverted index using $h$ in the same way as before, it now seems we have a strategy to solve the (approximate) nearest neighbor problem more efficiently: given a query $q$, only search for nearest neighbors in the bucket $h(q)$, since each of these points $x$ likely has $d(q, x) < r_1$. However, this strategy has a problem: with our definition, even a close neighbor might not be a collision with probability $(1 - p_1)$. Thus, we can repeat our inverted index $L$ times with different $h_i$ drawn independently from $H$, such that our probability of not finding a close neighbor in *any* bucket is $(1 - p_1)^L$. FALCONN [2] is an LSH inverted index algorithm that uses this basic idea, along with concatenation and probing tricks, to achieve an asymptotically optimal (and data-dependent sub-linear) runtime; see the paper and associated code repository for more details.

One final note is that in practice, most LSH families satisfy a much stronger condition than the above. Consider a similarity function $\text{sim} \in [0, 1]$, where $s(x, y) = 1 \implies x = y$. As $x$ and $y$ get more dissimilar (e.g. their distance increases according to some distance metric), $s(x, y)$ decreases. For most LSH families, there exists an explicit similarity function that their collision probability satisfies, such that $p(h(x) = h(y)) = \text{sim}(x, y)$. Such LSH families exist for most common similarity

functions, including cosine similarity (signed random projections) [8], Euclidean similarity ($p$-stable projections) [11], and Jaccard similarity (minhash or simhash) [6]. Following [13, 9, 10, 27, 24, 31], in our work, we use LSH families with this explicit similarity description to provide tight analyses and strong guarantees for similarity-search algorithms.

