# OpenReview forum: "DESSERT: An Efficient Algorithm for Vector Set Search with Vector Set Queries"
_NeurIPS.cc/2023/Conference — NeurIPS 2023 poster_

### Official Review · Reviewer_PFCR · 2023-07-07

**Soundness:** 3 good
**Presentation:** 3 good
**Contribution:** 3 good
**Rating:** 6
**Confidence:** 3

**Summary:**

The authors study the general case of multi-vector retrieval, i.e., ColBERT and beyond. They propose and analyze a new algorithm for this "vector set" search task, with theoretical guarantees. When integrated into ColBERT, the proposed DESSERT method is 2-5x faster at some, relatively small loss in quality.

**Strengths:**

1. The authors discuss a general class of "vector set" retrieval scoring functions (Sec 1.1), generalizing ColBERT.

2. The authors formalize the "vector set" search problem, perhaps for the first time (although I'd be interested in a comparison with Luan et al. 2021 "Sparse, Dense, and Attentional Representations for Text Retrieval" for completeness). They propose a new algorithm, unlike ColBERT's. The algorithm provides theoretical guarantees.

3. The proposed method is shown to be ~2pt worse than a recent state-of-the-art ColBERT method (PLAID) on MRR quality, while being 3x faster, applying search in just 15.5 milliseconds per query on CPU. This is on the Pareto frontier with respect to the ColBERT retrieval quality/latency curve.

**Weaknesses:**

DESSERT is primarily tested on the MS MARCO query set (besides a small synthetic experiment). The authors justify much of this by citing PLAID [35], but that other paper like many IR papers in the last 1.5-2 years tests on several datasets, including especially out-of-domain and larger datasets. It's not inherently clear how much quality loss DESSERT would incur out-of-domain, say, on the LoTTE or BEIR (or even a subset of one of them, if computational resources do not permit more tests). This is the main weakness in my opinion.

**Questions:**

- How much space does the full method consume? I see 15GB are consumed by the hash table. Is that all?
- How long does preparing the search data structures (indexing) take?

**Limitations:**

See weaknesses.

---

> ### Author Rebuttal · Authors · 2023-08-09
>
> *>DESSERT is primarily tested on the MS MARCO query set (besides a small synthetic experiment). The authors justify much of this by citing PLAID [35], but that other paper like many IR papers in the last 1.5-2 years tests on several datasets, including especially out-of-domain and larger datasets. It's not inherently clear how much quality loss DESSERT would incur out-of-domain, say, on the LoTTE or BEIR (or even a subset of one of them, if computational resources do not permit more tests). This is the main weakness in my opinion.*
>
> Thank you for the suggestion. We have run additional experiments on the LoTTE dataset. We find that DESSERT performs well in the out-of-domain setting. See the rebuttal PDF for full pareto tradeoffs on ten of the LoTTE datasets.
> Thank you for the reference to the work of Luan et al. We will take a deeper look and add this to our discussion of related work.
>
> __Questions:__
>
> __Q1: How much space does the full method consume? I see 15GB are consumed by the hash table. Is that all?__
>
> Yes, 15 GB is the space required by the full method. In general, DESSERT tables are an order of magnitude smaller than the (non-quantized) full embedding collection. For example, in our LoTTE experiments, the vast majority of configurations were under 4 GB.
>
> __Q2: How long does preparing the search data structures (indexing) take?__
>
> This is highly hardware-dependent. If a GPU (or accelerator) is available to cluster the centroids (same process used by PLAID), then DESSERT runs in a fraction of the embedding time. For example, MS-MARCO took several hours to embed on GPU, and then 2.5 additional hours to build the DESSERT index. We conducted a new set of experiments on LoTTE (using different CPU hardware), and here DESSERT took about 15 minutes to index 1.5M sets. The key takeaway is that DESSERT indexing adds only a small overhead to the existing embedding costs.
>
> If our response has addressed the issues raised in your review, we hope that you will consider raising the score.

---

> > ### Comment · Reviewer_PFCR · 2023-08-17
> >
> > Thank you for the response. I might be open to raising my score +1 to a 7 (accept) but I think the work needs a more crisp statement of where it's uniquely favorable compared to PLAID ColBERTv2 or ColBERTer. I can already see that the index is quite small (15GB) and that the method is quite controllable to trade away some quality for latency, in a graceful way.
> >
> > Would DESSERT in principle work well at 10x or 50x the MS MARCO current scale? Would you expect it to scale better or the same as other methods? How would latency vs storage scale?

---

> > > ### Author Response · Authors · 2023-08-19
> > > **Concise Summary of DESSERT Advantages**
> > >
> > > Thank you for the suggestion. We agree that it would be helpful to summarize the unique advantages of our method over other alternatives in a clear, concise way. Please see our proposed statement below:
> > >
> > > The primary advantage of DESSERT over previously proposed methods such as PLAID and ColBERTer is improved latency. DESSERT requires less time than competing methods to compare each query-target set pair. In practice, a primary driver of performance is the fact that DESSERT performs set-to-set comparisons with binary/integer operations as opposed to floating point multiplications/inner products. Thus, the latency advantages of DESSERT will persist even as we scale 10x or 50x. Finally, since the DESSERT index is low-memory (each set representation is smaller than the corresponding representation in ColBERT/PLAID), we can scale by an order-of-magnitude while staying within the RAM budget of a typical workstation.

---

### Official Review · Reviewer_iVRP · 2023-07-07

**Soundness:** 3 good
**Presentation:** 3 good
**Contribution:** 3 good
**Rating:** 6
**Confidence:** 4

**Summary:**

In this paper, the authors studied a new problem of vector set search with vector set queries. They have formalized this problem and proposed a novel, provable hashing scheme, DESSERT, to efficiently deal with this problem. Moreover, they have provided theoretical analysis for DESSERT and conducted experiments to validate its efficiency and accuracy.

**Strengths:**

- The motivation to study this new problem is clear, which is a vital sub-problem in semantic search, and the authors formalize a general definition of this problem.

- They design an efficient hashing to solve this challenging problem and provide a theoretical bound and query time complexity.

- The presentation is clear and easy to follow.

**Weaknesses:**

***Novelty and Contributions***

- I appreciate the authors have provided the theoretical bound and query time complexity for their proposed hashing scheme. One of my concerns is that the improvement of the query time complexity is only marginal compared to the brute force method, which is still O(N) in general. Moreover, the impact of L is less discussed either in the theory part or in the experiments, making the efficiency improvement less promising.

***Experiments***

- Another primary concern is the experiments. The experimental results are too few. They only validate the performance of DESSERT on a single real-world dataset, making their conclusion of DESSERT compared to PLAID less convincing. Moreover, no ablation study and parameter study is performed, making users hard to know the effectiveness of implementation tricks and how to set the hyperparameters of DESSERT. Please see Q1-Q4.

***Presentation***

- The paper still needs some careful proofreading. There are some minor issues in Algorithm 2. Please refer to Q5 for more details.

**Questions:**

***Experiments***

Q1: For the passage retrieval experiments, it is less convincing to me to use a single dataset for validation. Can they conduct and report results on at least one more dataset (for example, TREC Complex Answer Retrieval dataset [1] used in ColBERT)?

Q2: The metric R@1000 is too weak as an accuracy measure. Can they report recall together with R@1000?

Q3: What are the best hyperparameter settings of C, L, probe, and k for the results they show in Table 1? Moreover, can they conduct the parameter study for some vital parameters (e.g., L and C) of DESSERT?

Q4: In Section 5, the authors developed some implementation tricks for DESSERT. Can they conduct an ablation study to validate the effectiveness of the three tricks they proposed (i.e., filtering by k-means clustering, space optimized sketches, and the concatenation trick)?

***Presentations***

Q5: There exist some typos in Algorithm 2:
- In line 2: $G\circ H(G,S_i)$ -> $A_1 \circ A_2(Q,S_i)$;
- In line 4: $f_k(q)$ -> $f_L(q)$;
- In line 11: $count_{m_q}/L$ -> ${count}_{m_i}/L$;
- In line 13: $0,\cdots,N-1$ -> $1,\cdots,N$.

***Reference***

[1] Laura Dietz, Manisha Verma, Filip Radlinski, and Nick Craswell. 2017. TREC
Complex Answer Retrieval Overview. In TREC.

**Limitations:**

This work does not appear to have any negative social impact.

---

> ### Author Rebuttal · Authors · 2023-08-09
>
> *>I appreciate the authors have provided the theoretical bound and query time complexity for their proposed hashing scheme. One of my concerns is that the improvement of the query time complexity is only marginal compared to the brute force method, which is still O(N) in general. Moreover, the impact of L is less discussed either in the theory part or in the experiments, making the efficiency improvement less promising.*
>
> Thank you for pointing this out. In the revision, we plan to include better descriptions of these parameters (see also our response to Reviewer hUSH). In short:
> L is the number of hash functions used to approximate the similarity. L is a design parameter - higher values of L require a larger index size and query time but result in a smaller error when estimating the similarity scores. This can be seen from Lemma 4.2.3: for a fixed failure probability $\delta$, the estimation error decays with $\frac{1}{\sqrt{L}}$.
>
> Regarding the asymptotic complexity: This is true, but we want to emphasize that the asymptotic comparison does not fully capture the advantages of our method. The brute force method requires O(N) distance calculations, which typically require many floating-point multiplications. The most expensive asymptotic term in our method comes from integer comparisons and increments, which are substantially faster on hardware.
>
> __Experiments__
>
> We conducted a hyperparameter study, which we described on Line 301. We have added a more complete description of this study to the appendix. We have also added an evaluation on ten more datasets from the LoTTE benchmark.
>
> __Questions__
>
> __Q1: For the passage retrieval experiments, it is less convincing to me to use a single dataset for validation. Can they conduct and report results on at least one more dataset (for example, TREC Complex Answer Retrieval dataset [1] used in ColBERT)?__
>
> We have added an evaluation on ten of the LoTTE datasets, used in ColBERTv2. Our method shows extremely competitive Pareto-frontier performance. See the rebuttal PDF for full pareto tradeoffs on ten of the LoTTE datasets.
>
> __Q2: The metric R@1000 is too weak as an accuracy measure. Can they report recall together with R@1000?__
>
> R@1000 is the standard evaluation metric for the MS-MARCO evaluation task. However, we also report MRR@10 (see Table 1), which is a much harder metric to improve.
> In our LoTTE experiments, we report a variety of metrics (from R1@10 to R1@1000) using the evaluation code from the original paper.
>
> __Q3: What are the best hyperparameter settings of C, L, probe, and k for the results they show in Table 1? Moreover, can they conduct the parameter study for some vital parameters (e.g., L and C) of DESSERT?__
>
> We conducted this study, via grid search over the values described in the paper (see Line 301). The results of this study are listed below. We have added this to the appendix.
>
> Hyperparameter settings:
>
> Settings for DESSERT corresponding to the first row in table optimized for returning 10 docs:
> - Hashes_per_table (C) = 7
>
> - Num_tables (L) = 32
>
> - initial_filter_k = 4096
>
> - nprobe_query = 1
>
> Second row:
> - hashes_per_table  = 7
>
> - num_tables = 64
>
> - initial_filter_k = 4096
>
> - nprobe_query = 2
>
> Settings for DESSERT corresponding to the first row in table optimized for returning 1000 docs
> - hashes_per_table = 6
>
> - num_tables = 32
>
> - initial_filter_k = 8192
>
> - nprobe_query = 4
>
> Second row:
> - hashes_per_table = 7
>
> - num_tables = 32
>
> - initial_filter_k = 16384
>
> - nprobe_query = 4
>
> Intuitively, these parameter settings make sense: increase k and the number of total hashes for higher accuracy, and increase k for returning more documents (1000 vs. 10).
>
> __Q4: In Section 5, the authors developed some implementation tricks for DESSERT. Can they conduct an ablation study to validate the effectiveness of the three tricks they proposed (i.e., filtering by k-means clustering, space optimized sketches, and the concatenation trick)?__
>
> Unfortunately, all of these tricks are strictly required to have a practical index. Without the pre-filtering, the indexing process requires > 1 day to run. The concatenation trick provides a similar speedup. Finally, the process runs out of memory if we do not use space-optimized sketches, so this evaluation is not possible. We will make a note of this in the paper.
>
> If our response has addressed the issues raised in your review, we hope that you will consider raising the score.

---

> > ### Comment · Reviewer_iVRP · 2023-08-20
> > **Reply to Rebuttal**
> >
> > Thank you for the rebuttal.
> >
> > The authors have addressed all of my concerns. I am happy to see that much more datasets have been included in this manuscript, and the results make this work more convincing. I will raise my score to 6. Thanks!

---

### Official Review · Reviewer_FWNJ · 2023-07-09

**Soundness:** 3 good
**Presentation:** 2 fair
**Contribution:** 2 fair
**Rating:** 5
**Confidence:** 3

**Summary:**

The paper studies the vector set search problem, which is an extension of the canonical near-neighbor search problem and finds an application in the semantic search task. The paper claims that existing methods for vector set search are unacceptably slow, and thus, proposes an approximate search algorithm called DESSERT. The paper presents both theoretical analysis and empirical evaluation to demonstrate the effectiveness of the DESSERT algorithm.

**Strengths:**

S1. The paper proposes a new algorithm, DESSERT, to solve the vector set search problem efficiently.

S2. The paper presents both theoretical analysis and empirical evaluation to testify the effectiveness of the DESSERT algorithm.

S3. The DESSERT algorithm can be applied to the semantic search task.

**Weaknesses:**

W1. [Motivations & Contributions] The paper is the first to formally formulate the vector set search problem (if not, then citations are required to be provided to the first work), yet only providing one concrete application of the vector set search problem (i.e., semnatic search). The reasons why we need to study the vector set search problem require further clarified. It seems to me that the DESSERT algorithm is actually an incremental work, with a specific focus for improving the running time of the ColBERT model. This severely limits the paper's contributions.

W2. [Experiments] According to the paper, the proposed algorithm, DESSERT, is implemented in C++. However, the baseline method, the brute force algorithm, is implemented in Python using the PyTorch library. The reasonability and fairness of implementing two methods in different programming languages need further explanations.

W3. [Presentations] The paper's presentation needs improvement. There exist several typos and undefined notations in the paper. Some concrete examples are given as follows.

- Line 139: I believe the notation H refers to the hash set, which requires further clarification.

- Line 30: citations are required to the literature on traditional single-vector near-neighbor search.

- Algorithm 2: the output given in the pseudo code doesn't concur with the desciptions given in the main text (though they may refer to the same variable).



---
After the rebuttal process, most of my concerns mentioned above have been sufficiently addressed. Thus, I raise my rating from 4 to 5.

**Questions:**

Please refer to the Weaknesses given above, and provide further explanations, particularly for W1 and W2.

**Limitations:**

I do not see any potential negative societal impact in this paper.

---

> ### Author Rebuttal · Authors · 2023-08-09
>
> W1: Yes, we believe that we are the first to formalize the vector set search problem. While semantic search is the clear “killer application” today, we provide references to other potential applications such as database lineage tracking (line 96), image instance retrieval, market basket analysis, and graph neural networks (line 342). We focus on semantic search in our empirical evaluations because it is a core routine for various tasks (web search, product search, recommendation ranking, and question-answering). We hope that our general formulation of the problem will encourage future work in this area, especially since we have shown that search is tractable for a large class of similarity functions.
>
> Secondly, we respectfully disagree with the claim that accelerating ColBERT is an incremental contribution. ColBERT has had a tremendous impact on both research and practice. The “late interaction” modeling approach introduced by ColBERT represents the state of the art in numerous IR benchmarks such as passage retrieval [1], multi-hop reasoning [2], open question answering [3], and even broader retrieval problems such as table QA [4]. Given the widespread interest and adoption of ColBERT and the importance of achieving low latency inference on cost-effective hardware in industry settings [5] , we believe that our work has the potential for substantial real-world impact.
>
> [1] https://arxiv.org/abs/2205.09707
>
> [2] https://arxiv.org/abs/2101.00436
>
> [3] https://arxiv.org/abs/2007.00814
>
> [4] https://aclanthology.org/2023.acl-short.133/
>
> [5] https://arxiv.org/abs/2101.09086
>
>
> W2: PyTorch is not a pure python library and heavily calls out to optimized C++ and CUDA kernels for tensor operations. Similarly, we implement DESSERT with a C++ backend and invoke the algorithm via a python interface (please see our code provided in the supplementary materials for further details). Thus, this is an apples-to-apples comparison because both PyTorch and our codebase are Python binding-based C++ libraries.
>
> W3:
>
> - Line 139: This is the set of functions from which we draw the LSH function. It is defined in line 126.
>
> - We discuss this literature in lines 83-94. We will also add the citations in line 30
>
> - We have cleaned up the algorithm pseudocode description by including a table for notation as suggested by Reviewer hUSH. We also fixed multiple typos in the algorithm presentation.
>
> If this has addressed the issues in your review, we hope that you will consider raising the score.

---

### Official Review · Reviewer_Uhsc · 2023-07-13

**Soundness:** 4 excellent
**Presentation:** 4 excellent
**Contribution:** 3 good
**Rating:** 7
**Confidence:** 2

**Summary:**

The paper addresses the problem of set-to-set similarity calculation and retrieval, which is a problem with any downstream applications. While previous approaches inevitably perform a brute force similarity calculation over |Q| query vectors and |S| target vectors for each set comparison F(Q, S), this paper leverages random hashing functions to estimate similarity between query and target vectors to avoid this brute force search. Combined with highly space-optimized hash tables, and building on top of previous centroid-based candidate set filtering mechanisms, the proposed algorithm achieves notable speed improvements over existing methods at only a small cost to recall. Theoretical support is provided for the algorithm, along with experiments on synthetic, solvable data and the standard MS MARCO benchmark dataset.

**Strengths:**

- Very important: The proposed method cleverly utilizes randomness via an LSH based approach to estimate similarity and avoid brute force similarity calculation between sets of vectors, leading to notable improvements in speed over previous methods, at reasonable memory requirements for large scale settings, and with very limited negative impact on recall.
- Important: The experiments are simple and clear.
- Important: The choice of baselines seems appropriate. As the authors note, their method reaches points on the Pareto curve, w.r.t. latency, that are not reachable by baselines even though the baselines are nominally customizable for trading off latency vs. recall.
- Important: The paper is clearly written.
- Of some importance: Theoretical analysis is provided for the proposed method, although, as the authors note, they “assume sufficiently high relevance scores and large gaps in our theoretical analysis to identify the correct results.”

**Weaknesses:**

- Important: As noted by the authors, this work draws heavily upon LSH-based NN vector search methods that already use LSH methods to approximate exact similarity calculation. In my opinion, the extension to the set-to-set comparison setting is fairly straightforward. This is a drawback of the methodological contribution of the paper, in terms of novelty, though it may be entirely outweighed by the positive empirical performance of the approach over strong baselines.
- Of some importance: The intro of the paper is written as if the paper proposes a generic framework for cross-query-vector aggregation (A2) and cross-target-vector aggregation steps (A1), but to my understanding, the only setting experimentally considered in the paper is that of A2 = weighted sum and A1 = max. This seems to be the main practical setting of interest in settings like passage retrieval, but I would not say this paper really broadens how people have been thinking about set-to-set comparison.

**Questions:**

- The paper argues that vector-level NN retrieval is clearly limited, but I think this assumes that one is really looking for a comparison using something like A2=weighted sum and A1=max. I think there are settings where one would be interested in A2=max and A1=max, as well as other variants. Are there problems besides passage retrieval that are better evaluated in terms of other combinations of A2 and A1 besides those considered in this paper?
- The set-to-set comparison problem might also be viewed more generally as comparison of two distributions over R^d. When |Q| and |S| are large relative to d, these distributions might be accurately estimated and then distributional divergence measures deployed. What do you think the trade-offs are with a vector similarity-based approach? (And could the perspectives be unified based on specifying A2/A1 and sim?)
- I think I never saw which specific LSH functions you used, besides using 16/32/64 of them. Are they all signed random projection to approximate cos-sim?
- What does perfect recall mean on synthetic data? That you already know what the correct top-1 avg-max-sim “passage” is?
- Can you increase the latency of your approach to reach the performance of PLAID?
- In the next version of the appendix, could you include the results of your hyperparameter grid search?
- Since you make a claim about Pareto improvement, it would be nice to include graphs visualizing the Pareto curves of the methods.
- More of a comment, really: You might add “passage retrieval” for language modeling pretraining and finetuning to the list of potentially valuable applications for this kind of approach. Current methods use simple heuristics like embedding averaging across tokens to reduce the problem to vector-level NN search: https://arxiv.org/pdf/2112.04426.pdf.

**Limitations:**

Yes, the limitation section was satisfactory.

---

> ### Author Rebuttal · Authors · 2023-08-09
>
> Regarding novelty: While LSH methods have been widely used for the last 20 years, our set search departs from the standard approach in some subtle but important ways. The vast majority of LSH search algorithms group points into hash tables, then explicitly compute similarities between points that fall into the same buckets. A naive adaptation of this method would actually require more memory than PLAID. Instead, we use collision statistics to rank sets based only on the distribution of hash codes, meaning that we do not store the d-dimensional embeddings. While similar ideas have recently been used for vector similarity search [1] and density estimation [2], this is still an emerging area and a new interpretation of LSH.
> While it is true that the NLP community currently only uses A2 = (unweighted) sum and A1 = max, our theory shows that the set-search problem is tractable for a larger family of functions. We hope that this might lead to better modeling techniques (e.g., learned weights in the sum, or soft nonlinearities in place of the maximum), though this is beyond the scope of our present study. Our general analysis provides some insight about which kinds of functions might be good candidates (e.g., Lemma 4.2.1 identifies smooth, monotone functions as compatible with fast search).
>
> __Questions:__
>
> 1. While it is true that the NLP community currently only uses A2 = (unweighted) sum and A1 = max, our theory shows that the set-search problem is tractable for a larger family of functions. We hope that this might lead to better modeling techniques (e.g., learned weights in the sum, or soft nonlinearities in place of the maximum), though this is beyond the scope of our present study.
> Problems such as database lineage tracking [1] use a linear combination of average and max similarity (and [1] references papers that use different A1 / A2). Image instance retrieval can be formulated as A1=max and A2=max (e.g., over SIFT vectors), but again other compositions are possible [2]. We think that the design of A1 and A2 represents an exciting opportunity for future work, since our results show that a large class of similarity aggregation functions can be solved tractably.
>
> 	[1] https://arxiv.org/pdf/2107.06817.pdf
>
> 	[2] https://arxiv.org/pdf/2101.11282.pdf
>
>
> 2. This idea is very interesting. For divergence-based ranking to be feasible, we would need a fast, low-memory way to model high-dimensional distributions and compute divergences. Most methods scale poorly with input size and dimensionality (in both runtime and error), but some recent LSH-based algorithms [2, 3] can construct a kernel density estimate of the distribution in near-linear time. If a divergence measure can be constructed from these methods, it would lead to a tradeoff where DESSERT requires $O(m m_q)$ time to materialize the pairwise similarity matrix, but the divergence calculation runs in time $\tilde{O}(m + m_q + C)$ where $C$ is the cost to compute the divergence.
>
> For example, consider the kernel density estimate of the total variation distance.
> $$\int_z \left|\sum_{q \in Q}k(q, z) - \sum_{x \in S}k(x, z)\right|dz$$
> If we approximate this integral by sampling point estimates at a set $Z$ of sampling locations (e.g. Monte Carlo estimation), we have an estimate of the form:
> $$\frac{c}{|Z|}\sum_{z \in Z} \left|\sum_{q \in Q}k(q, z) - \sum_{x \in S}k(x, z)\right|dz$$
> where $c$ is a scalar. If the quantity $k(x, z)$ is chosen from the family of LSH similarity kernels [2], it can be approximated via DESSERT if we choose $z$ from the set $Q$. This gives the decomposition:
> $$A_1(A_2(q, S)) = c\sum_{z \in Q}\left| \sum_{z \in Q} k(q,z) - A_2(z, S) \right|$$
> $$A_2(q,S) = \sum_{x \in S} \sigma(q, x)$$
> $$\sigma(x,z) = k(x,z)$$
> Unfortunately, it is not clear whether the restriction $z \in Q$ ruins the Monte Carlo estimator. It is also not clear whether other divergences (such as the KL divergence or Hellinger distance) can be similarly decomposed. Another problem is that nonparametric density estimation suffers from the curse of dimensionality: $|S|$ must grow much faster than $d$ to have constant estimation error (so this may only work for very large sets). This is a fascinating direction for future work.
>
> References:
>
> [1]: “FLASH: Randomized Algorithms Accelerated over CPU-GPU for Ultra-High Dimensional Similarity Search,” Wang et. al., SIGMOD 2018
>
> [2]: “Sub-linear RACE Sketches for Approximate Kernel Density Estimation on Streaming Data,” Coleman and Shrivastava, WWW 2020
>
> [3]: “Rehashing Kernel Evaluation in High Dimensions,” Siminelakis et. al., ICML 2019
>
> 3. Yes, the LSH functions are all signed random projections.
>
> 4. Yes, we computed the top-1 avg-max-sim “passage” to use as ground truth for the synthetic experiments.
>
> 5. Regarding whether DESSERT can match PLAID: We’ve conducted new experiments on LoTTE that show the full latency-performance tradeoff. Our experiments show that it is sometimes (but not always) possible to match the PLAID performance. This is probably due to the estimation error of our similarity approximation. Lemma 4.2.3 shows that in principle, this error can be driven arbitrarily small (as $L \to \infty$, the error between the DESSERT and PLAID ranking scores goes to zero with high probability). However, the error scales with $\frac{1}{\sqrt{L}}$, so zero error requires large values of $L$ that may not be feasible in practice.
>
> 6. We plan to include these results, as well as the Pareto-optimal hyperparameter configurations for LoTTE.
>
> 7. See our evaluation on LoTTE (pdf attached in the author rebuttal), where we show the full Pareto curves. On the plots, the PLAID result shows the lowest-latency result attainable by PLAID. We did not do the full curves for MS-MARCO due to computational constraints.
>
> 8. Thank you for the reference! We will mention this in the revision.
>
>
> If our response has addressed the concerns raised in your review, we hope you will consider raising the score.

---

> > ### Comment · Reviewer_Uhsc · 2023-08-15
> > **Reply to rebuttal**
> >
> > Thanks for the thorough reply. The response addresses all of my questions. Where my main concern was novelty, I appreciate that a naive extension of existing methods to the set-to-set setting would be extremely memory intensive, and the proposed algorithm makes some advances on that front. Other parts of the response point out ways that the paper theory and method are relatively general across possible applications, although the experiments in the paper remain somewhat narrow in their focus. I raise my score from 6 to 7 as a result, though I am keeping my confidence at 2 because I am not intimately familiar with the area for the paper.

---

### Official Review · Reviewer_hUSH · 2023-07-17

**Soundness:** 4 excellent
**Presentation:** 3 good
**Contribution:** 3 good
**Rating:** 7
**Confidence:** 4

**Summary:**

The paper presents a study focused on the problem of vector set search with vector set queries, which is a crucial subroutine for various web applications. The authors highlight the insufficiency of existing solutions in terms of speed. To address this, they propose a new approximate search algorithm called DESSERT. DESSERT is a versatile tool that offers robust theoretical guarantees and demonstrates excellent empirical performance. The authors integrate DESSERT into ColBERT, a highly optimized state-of-the-art semantic search method, and report a notable 2-5x speed improvement on the MSMarco passage ranking task.

**Strengths:**

1. The problem formulation is truly innovative. Vector set search is undeniably an essential and challenging task, with practical applicability in serving LLMs.

2. The proposed methodologies are lucid and firmly grounded, ensuring the reproducibility of the research.  DESSERT uses hash tables for element-wise similarity search to build a data structure for vector set similarity search. Moreover, DESSERT taking advantage of hash tables in terms of query speed and performs an efficient query for similar sets.

3. DESSERT has a detailed theoretical analysis in terms of inner and outer aggregation, which justify its choices of aggregation functions in the algorithm.


**Weaknesses:**

It would be better to have a section that unifies the notation for algorithms and theory. Maybe a table would be better.

Overall, I have a positive impression on the paper. If authors are willing to further polish the theory part of the paper, I'm willing to raise the score.

**Questions:**

1. How can DESSERT be applied to different similarity measures?

2. How to intuitively explain the running time and memory? How do the terms $L$  and $T$ relate to the search quality?

**Limitations:**

The authors adequately addressed the limitations.

---

> ### Author Rebuttal · Authors · 2023-08-09
>
> Thanks for your review and suggestions. We agree that the theory is a little hard to read, so we’ve added a table to disambiguate. We’ve reproduced (part of) the table below and have added this to the appendix. We have also updated the notation in the algorithm listings, since there were some mistakes (e.g., $k$ instead of $L$ as the hash subscript in line 4 of Algorithm 2). We also plan to standardize the notation between the practical and theoretical sections (e.g., we can rewrite Algorithm 2 to replace the “count” variable with $\hat{\mathbf{s}}(q, S_i)$). We thank you for pointing out these discrepancies and we plan to carefully consolidate the notation as much as we can.
>
> | Notation                     | Definition                                                | Explanation                                                                              |
> | ---------------------------- | --------------------------------------------------------- | ---------------------------------------------------------------------------------------- |
> | $D$                          | Set of target sets                                        | Collection of documents                                                                  |
> | $N$                          | Cardinality $\\vert D \\vert$                                         | Number of documents                                                                      |
> | $\\mathcal{D}$               | DESSERT index of $D$                                      | Search index data structure                                                              |
> | S                            | Target set                                                | Document (text passage)                                                                  |
> | Q                            | Query set                                                 | Search phrase (text passage)                                                             |
> | $x \\in S_i$                 | Vector in target set $S_i$                                | Embedding                                                                           |
> | $q \\in Q$                   | Vector in query set $Q$                                   | Embedding                                                                           |
> | $m_i$                        | Cardinality $\\vert S_i \\vert$                                       | Length of $i$th document                                                                 |                                                                   |                                      |
> | $\\mathrm{score}_i$          | Estimate of $F(Q,S_i)$                                    | Approximate relevance /  search ranking score                                            |
> | $L$                          | Number of hashes                                          | Larger $L$ increases the accuracy of $\\mathrm{score}$ at the cost of space and latency. |
>
>
> __Questions:__
>
> *Q1: How can DESSERT be applied to different similarity measures?*
>
> To use DESSERT with a similarity measure other than cosine similarity, we can use a different LSH function. There are well-known LSH functions for similarities based on Euclidean, Manhattan, and Lp-norm distances [1] as well as measures defined over sets and Hamming codes (such as Jaccard similarity [2-4], edit distance, etc). There is also a developing research area using asymmetric hashing techniques [6] to design more flexible, asymmetric similarity measures.
> [1]: “Locality-Sensitive Hashing Scheme Based on p-Stable Distributions,” Datar et. al. in SCG 2004.
> [2]: “One Permutation Hashing,” Li et. al. in NIPS 2012
> [3]: “Densifying One Permutation Hashing via Rotation for Fast Near Neighbor Search,” Shrivastava and Li, ICML 2014
> [4]: “Re-randomized Densification for One Permutation Hashing and Bin-wise Consistent Weighted Sampling
> [5]: “Locality-sensitive hashing for the edit distance,” Marcais et. al., Bioinformatics, Volume 35, Issue 14, July 2019
> [6]: “Asymmetric LSH (ALSH) for Sublinear Time Maximum Inner Product Search (MIPS),” Shrivastava and Li, NIPS 2014
>
> *Q2: How to intuitively explain the running time and memory? How do the terms L and T relate to the search quality?*
>
> T is the maximum number of elements in a single set that share the same hash code. This bound can be loosely interpreted as saying, “No set $S_i$ contains too many elements that are very similar to a single query vector $q \\in Q$.” In the text-search setting, T can be thought of as the number of duplicate words (or very close synonyms) in the document.
>
> To develop intuition for why we need this bound, look at lines 9-10 of Algorithm 2. If an element of the query set $Q$ collides with every element in a target set $S_i$, the loop in lines 9-10 takes time $|S_i|$. If this happens for every target set, the resulting algorithm is asymptotically no better than brute force (albeit with expensive distance calculations replaced by cheap integer operations). However, this also does not happen except for in highly degenerate, unrealistic cases where the document consists entirely of duplicate / near-duplicate terms (e.g., the document “cat cat cat kitten cat” and the query “cat kitten”). In practical search datasets, we’ve found that T is fairly small (usually under 20).
>
> L is the number of hash functions used to approximate the similarity. L is a design parameter - higher values of L require a larger index size and query time but result in a smaller error when estimating the similarity scores. This can be seen from Lemma 4.2.3: for a fixed failure probability $\delta$, the estimation error decays with $\frac{1}{\sqrt{L}}$.
> We will add this discussion to Section 4.3.
>
>
> If this rebuttal has addressed your concerns, we hope you may consider raising the score.

---

> > ### Comment · Reviewer_hUSH · 2023-08-10
> >
> > Thanks for your response! I like the notation table a lot and hope you can incorporate it in your revision. I'm also happy to have my questions answered and concerns addressed. Thus, I'll raise my score to 7.

---

### Official Review · Reviewer_yuRv · 2023-07-18

**Soundness:** 3 good
**Presentation:** 3 good
**Contribution:** 3 good
**Rating:** 7
**Confidence:** 3

**Summary:**

This paper considers a nearest neighbor search problem where each point is a set of vectors, and the distance function is drawn from a general class of aggregation functions over the vectors.

The approach presented here is based on the LSH algorithms, but since we're dealing with multiple vectors, the bounds behind LSH need to be re-derived. Assumptions used in order to allow or speed up the search include that the distance function obeys a certain Lipschitz smoothness property, and tools include inverted indices, sampling, and compressed tables. These are all fairly standard in the field.

**Strengths:**

The query model introduced is interesting, and the authors make a good argument as to its utility. The techniques utilized are non-trivial, and constitute a contribution to the field of nearest neighbor search. The fact that the distance function is very general is also a plus.

**Weaknesses:**

The presentation can be somewhat improved. The authors actually put in significant work into this direction, but the presentation can be improved by adding a high-level overview of the approach before introducing the pseudo-code. This is a more effective approach than referring to code to explain an approach.

The authors should also add background on LSH and inverted indices, so that the paper can be more accessible to non-experts.

The explanation of the motivation behind TinyTable is garbled, and a clearer exposition is necessary.

---------------------------------

In response to the author rebuttal, and in particular the improvement in presentation, I raised my score.

**Questions:**

None.

---

> ### Author Rebuttal · Authors · 2023-08-09
>
> Thank you for pointing out the presentation issues.
>
> Regarding high-level overview: Good suggestion. We’ve added the following description to Section 3: “At a high level, DESSERT compresses the collection of target sets into a form that makes similarity operations efficient to calculate. This is done by replacing each element of the set with its LSH codes, transforming the set into an integer array of hash values. At query time, these hash values are compared with the corresponding hashes of the query set elements to approximate the pairwise similarity matrix (Figure 1). This matrix is used as the input for aggregations $A_2$ and $A_1$ to rank the target sets.”
>
> Regarding background: We began with a longer text that contained a thorough exposition on LSH and inverted indices, which we had to cut down due to the NeurIPS page limit. In the revision, we have put this background back into the appendix and provided references in the main text for the interested reader.
>
> Regarding TinyTable: After a second read-through, we agree. We’ve drafted a revised description of TinyTables, which we hope will be clearer:
> “DESSERT has two features that constrain the underlying hash table implementation: (1) every document is represented by a hash table, so the tables must be low memory, and (2) each query performs many table lookups, so the lookup operation must be fast. If (1) is not met, then we cannot fit the index into memory. If (2) is not met, then the similarity approximation for the inner aggregation step will be far too slow. Initially, we tried a naive implementation of the table, backed by an std::vector, std::map, or std::unordered_map. In each case, the resulting structure did not meet our criteria, so we developed TinyTable, a compact hash table that optimizes memory usage while preserving fast access times. TinyTables sacrifice O(1) update-access (which DESSERT does not require) for a considerable improvement to (1) and (2).”
>
> If this rebuttal has addressed your concerns, we hope you may consider raising the score.

---

### Author Rebuttal · Authors · 2023-08-09

We thank all of the reviewers for their thoughtful feedback!

It seems the most common theme expressed in the reviews was a desire for evaluation on more datasets. To that end, we have run 10 additional evaluations on the LoTTE dataset for out-of-domain retrieval. Interestingly, we found that DESSERT performed very well compared to PLAID in this setting. We hope that this sufficiently addresses the concerns of those reviewers who desired to see an evaluation on more datasets.  We have attached a pdf plotting the pareto curves for DESSERT and PLAID on these datasets.

---

### Decision · Program_Chairs · 2023-09-21

**Decision:**

Accept (poster)

**Comment:**

The paper introduces DESSERT, a novel algorithm for efficient vector set search, supported by theoretical analysis and empirical validation. Moreover, DESSERT finds application in the realm of semantic search tasks. It employs hash tables for element-wise similarity search, enhancing query speed and overall efficiency. The reviewers are in favor of accepting the submission, but please incorporate all the reviewers' suggestions in the final version of the paper.